# IgCaller for reconstructing immunoglobulin gene rearrangements and oncogenic translocations from whole-genome sequencing in lymphoid neoplasms

Ferran Nadeu [1,2✉], Rut Mas-de-les-Valls[1], Alba Navarro[1,2], Romina Royo[3], Silvia Martín[1,2], Neus Villamor [1,2,4], Helena Suárez-Cisneros[5], Rosó Mares[1], Junyan Lu[6], Anna Enjuanes[1,5], Alfredo Rivas-Delgado [1,4], Marta Aymerich[1,2,4], Tycho Baumann[4], Dolors Colomer[1,2,4,7], Julio Delgado[1,2,4], Ryan D. Morin [8,9], Thorsten Zenz [10], Xose S. Puente [2,11], Peter J. Campbell [12], Sílvia Beà [1,2,7], Francesco Maura [12,13] & Elías Campo [1,2,4,7]

Immunoglobulin (Ig) gene rearrangements and oncogenic translocations are routinely assessed during the characterization of B cell neoplasms and stratification of patients with distinct clinical and biological features, with the assessment done using Sanger sequencing, targeted next-generation sequencing, or fluorescence in situ hybridization (FISH). Currently, a complete Ig characterization cannot be extracted from whole-genome sequencing (WGS) data due to the inherent complexity of the Ig loci. Here, we introduce IgCaller, an algorithm designed to fully characterize Ig gene rearrangements and oncogenic translocations from short-read WGS data. Using a cohort of 404 patients comprising different subtypes of B cell neoplasms, we demonstrate that IgCaller identifies both heavy and light chain rearrangements to provide additional information on their functionality, somatic mutational status, class switch recombination, and oncogenic Ig translocations. Our data thus support IgCaller to be a reliable alternative to Sanger sequencing and FISH for studying the genetic properties of the Ig loci.

[1] Institut d'Investigacions Biomèdiques August Pi i Sunyer (IDIBAPS), Barcelona, Spain. [2] Centro de Investigación Biomédica en Red de Cáncer (CIBERONC), Madrid, Spain. [3] Barcelona Supercomputing Center (BSC), Barcelona, Spain. [4] Hospital Clínic of Barcelona, Barcelona, Spain. [5] Unitat de Genòmica, IDIBAPS, Barcelona, Spain. [6] European Molecular Biology Laboratory (EMBL), Heidelberg, Germany. [7] Universitat de Barcelona, Barcelona, Spain. [8] Department of Molecular Biology and Biochemistry, Simon Fraser University, Burnaby, BC, Canada. [9] Canada's Michael Smith Genome Sciences Centre, British Columbia Cancer Agency, Vancouver, BC, Canada. [10] Department of Medical Oncology and Hematology, University Hospital and University of Zürich, Zürich, Switzerland. [11] Departamento de Bioquímica y Biología Molecular, Instituto Universitario de Oncología, Universidad de Oviedo, Oviedo, Spain. [12] Wellcome Sanger Institute, Hinxton, Cambridgeshire, UK. [13] Myeloma Service, Department of Medicine, Memorial Sloan Kettering Cancer Center, New York, NY, USA. ✉email: nadeu@clinic.cat

Mature normal and tumor B cells express a unique immunoglobulin (Ig) gene rearrangement. This individual Ig gene is formed during the first steps of B cell development in the bone marrow where both heavy (IGH) and light chains [kappa (IGK) or lambda (IGL)] are rearranged by a hierarchical process in which distant variable (V), diversity (D, only in the IGH locus), and joining (J) genes are joint through deletions of the genomic sequence between them[1]. During this process of cut-and-joining of the different genes, random nucleotides known as N nucleotides are added in the junctions to increase the diversity of the Ig repertoire. Later on, upon antigen activation, the Ig gene rearrangements undergo further diversification by the process of somatic hypermutation (SHM), which introduces mutations in the V(D)J regions, and class switching of the heavy chain in the germinal center of the lymphoid follicles[1].

The identification of a clonal B cell population (i.e. large B cell population expressing the same Ig) in the context of a lymphoid proliferation is used as a marker of leukemia/lymphoma diagnosis. Besides, the presence of SHM in the V(D)J region of the IGH is a surrogate imprint of the cell of origin of the lymphoid neoplasm with marked clinical implications. In chronic lymphocytic leukemia (CLL)[2,3] and mantle cell lymphoma (MCL)[4], the identification of SHM distinguishes subtypes of tumors [mutated (M-IGHV) or unmutated (U-IGHV)] with different clinical and biological behavior. In CLL, different prognostic models incorporate the IGHV mutational status[5], and guidelines recommend its analysis either at diagnosis or before treatment initiation[6]. In addition, tumors carrying highly similar Ig (i.e. stereotyped Ig)[7], specific Ig light chain (IGLC) rearrangements[8], or class switch recombination (CSR) of the constant region of the heavy chain[9] characterize subsets of patients with distinct clinical and biological features. Besides, the detection of oncogenic Ig translocations genome-wide helps the diagnosis of different neoplasms such as MCL, follicular lymphoma and Burkitt lymphoma, while stratifies patients with distinct clinical outcomes in multiple myeloma (MM) and diffuse large B cell lymphoma (DLBCL)[10].

The analysis of the rearranged Ig gene is currently performed both in the clinical routine and research by Sanger sequencing (SSeq)[6] or by specific targeted next-generation sequencing (NGS) protocols[11]. Of note, independent assays are required to assess the IGH, IGK and IGL sequences[12]. Once the rearranged sequence is obtained, several tools are available to identify the V(D)J genes, its functionality (i.e. productive or unproductive) and mutational status (IMGT/V-QUEST[13,14] or IgBLAST[15]), and stereotype (ARResT/AssignSubsets[16]). On the other hand, Ig translocations are routinely assessed by fluorescence in situ hybridization (FISH) and/or conventional cytogenetics in the clinical setting. Although short-read whole-genome sequencing (WGS) of B cell neoplasms should store the information to reconstruct the entire Ig gene, the high genomic complexity of the Ig loci has prevented its analysis using the current bioinformatic pipelines. The decreasing cost of short-read WGS linked with its ability to characterize the entire genomic landscape of these neoplasms in a single experiment[17], even if complex and heterogeneous[18], suggests that WGS could enter into the clinical setting in the near future.

Here we present IgCaller, a fast, easy-to-run, python program designed to reconstruct the entire Ig gene rearrangements from short-read WGS data of lymphoid neoplasms. We demonstrate the accuracy of IgCaller using WGS data of 404 B cell neoplasms with available SSeq/NGS of the IGH V(D)J and/or IGLC and isotype expression for comparison: 230 cases of CLL in two independent cohorts of 152 (cohort 1 [C1])[18,19] and 78 (cohort 2 [C2]), 64 cases of MCL[20], 30 MM[21], 73 DLBCL[22], and 7 mature B cell non-Hodgkin lymphomas (B-NHL) (Supplementary Data 1).

## Results

**Overview of IgCaller.** IgCaller takes as input the WGS aligned reads (BAM file)[23–25] to assemble the rearranged IGH V(D)J genes, IGK and IGL VJ genes, and to identify the presence of CSR and genome-wide Ig translocations. IgCaller also determines the identity of the rearranged sequences compared to the germ line of the patients or reference genome. Although IgCaller also produces a preliminary analysis of the functionality of the rearranged sequences, these sequences can be used as input of downstream programs such as IMGT/V-QUEST or IgBLAST, as usually done for the sequences obtained from SSeq/NGS (Fig. 1a).

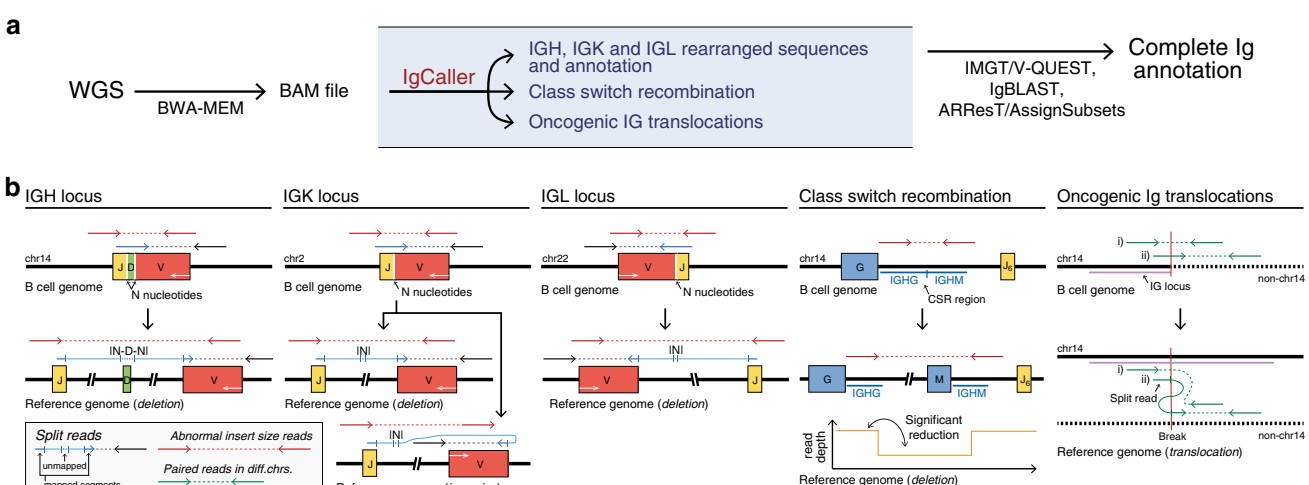

**Fig. 1 Overview of IgCaller and Ig loci at short-read WGS level. a** Bioinformatic steps to fully characterize the rearranged Ig gene from short-read WGS data. IgCaller extracts the rearranged sequences from already aligned reads (BAM file). The output of IgCaller might be used as input to downstream specific programs for a complete Ig annotation. **b** Schema of the Ig loci at B cell genome level (top) and reference genome level (bottom) for the different loci/rearrangements analyzed by IgCaller. Of note, although the IGK locus is oriented on the negative strand, the IGKV4-1, IGKV5-2, and IGKV genes within the distal cluster are inverted and therefore oriented on the positive strand. Thus, rearrangements involving these IGKV genes are formed through inversions rather than deletions. The WGS reads that cover the rearrangements and are used by IgCaller are depicted in each scenario. White arrows represent the coding strand.

IgCaller relies on two sets of reads to identify the break points at base pair resolution and to reconstruct the rearranged V(D)J sequences (Fig. 1b): (i) split reads (a fraction of the read maps to one location while the other fraction of the same read maps to a different location of the genome) spanning the boundaries of both V and J genes [note that in the IGH loci, reads do no map to the D gene due to its small length]; and (ii) abnormal insert size reads (reads with anomalous distance between read pairs) in which one read of a pair maps to a V gene and the other to a J gene. Once the rearranged V–J gene is found by combining both sets of reads, the consensus unmapped sequence of the split reads is used to extract the sequence containing the N nucleotides-D gene-N nucleotides (IGH locus) or the N nucleotides (IGK/IGL loci). The germ line sequence of the patient is used, if available, to consider potential polymorphisms when assessing the identity of the rearranged sequence to the germ line. The read depth before and after the potential CSR identified is compared to determine the presence of isotype switching[26]. Besides, using both types of reads as well as paired reads aligning to different chromosomes, IgCaller identifies genome-wide rearrangements (deletions, inversions, gains and translocations) involving any of the Ig loci. A detailed explanation of the methodological framework and instructions to run IgCaller might be found in the "Methods".

**Ig heavy chain rearrangements and identity**. Using the WGS of 404 B cell neoplasms, IgCaller identified a complete productive IGH gene rearrangement [V(D)J] in 131 (86%) C1-CLL, 75 (96%) C2-CLL, 63 (98%) MCL, 21 (70%) MM, 44 (60%) DLBCL, and in all 8 B-NHL. A partial (VJ) rearrangement was detected in 10 (7%) C1-CLL and 1 (3%) MM (Fig. 2a, Supplementary Data 2–7). Two distinct productive IGHV-IGHD-IGHJ gene rearrangements

were observed in 2 cases (1 CLL, 1 MCL). These results were fully concordant with the SSeq/NGS of 139 C1-CLL, 67 C2-CLL, 60 MCL, and 1 B-NHL (Supplementary Data 2–7). Small discrepancies (only J or V disagreement) were found when the J ($n = 7$) or V ($n = 1$) genes identified by SSeq based on identity (IMGT/V-QUEST) did not correspond to the rearranged genes detected by IgCaller, but were the second scoring genes in IMGT/V-QUEST, suggesting that our non-identity WGS-based approach might be more accurate in these scenarios (Supplementary Fig. 1). Contrarily, rearrangements within genes not annotated in the reference genome used by IgCaller could lead to incongruent results. These errors occurred in five CLL patients carrying rearrangements involving IGHV5-10-1 ($n = 4$) or IGHV7-4-1 ($n = 1$), which are not annotated in the hg19 genome build used. All these rearrangements were recovered after aligning the WGS data to the hg38 reference genome (Supplementary Fig. 2). Of note, the sequence of the complete IGH gene rearrangement identified by IgCaller could be used to determine the stereotypy of the CLL cases (Supplementary Data 2 and 3). IgCaller also reports the unproductive rearrangements. In this regard, IGH unproductive rearrangements were identified in 51 cases, all but four carrying productive rearrangements in the other allele (Supplementary Data 8). We verified by NGS 7 randomly selected IGH unproductive rearrangements (Supplementary Data 8).

Next, the comparison of the percentage of identity of the rearranged sequence to the germ line in 139 C1-CLL, 67 C2-CLL, and 60 MCL obtained by SSeq/NGS and IgCaller showed a high significant correlation and concordance in all three cohorts (Fig. 2b, Supplementary Fig. 3). Only 2 (0.8%) cases with a complete rearrangement and a partial rearrangement by WGS,

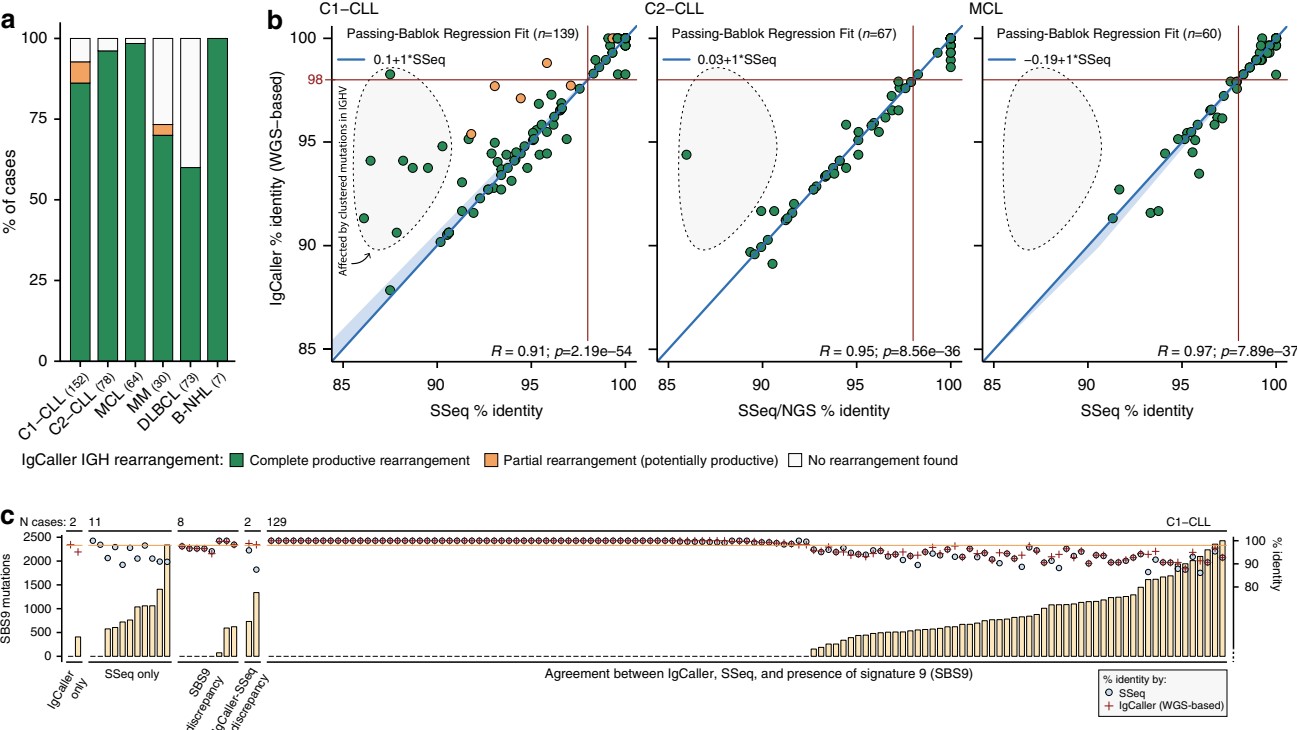

**Fig. 2 Benchmarking of IgCaller: characterization of the IGH locus. a** Bar plot showing the percentage of cases with productive IGH rearrangements by IgCaller in each cohort. **b** Dot plots of the percentage of identity of the rearranged IGHV sequence to the germ line by IgCaller (y axis) and SSeq/NGS (x axis). The 95% confidence interval is depicted by the light blue area. The gray area highlights cases in which the presence of a high density of clustered mutations impairs an accurate identification of the percentage of identity. P values are from t-test. **c** Comparison of the number of mutations associated with signature 9 (SBS9, left y axis) and the identity of the rearranged sequence both by SSeq and IgCaller (right y axis) in the C1-CLL cohort. Source data are provided as a Source data file.

respectively, were differentially classified as M-IGHV or U-IGHV between SSeq and WGS using the standard cut off of 98%. In line with this, we observed that WGS reads carrying a high density of clustered mutations might not align, and therefore the identity of the rearranged sequence could be overestimated (Supplementary Fig. 4). However, this non-alignment of highly mutated WGS reads only affected a minority of CLL cases, and rarely miss-classified patients as M-IGHV or U-IGHV (Fig. 2b). Besides, the use of WGS allows the recognition of the germinal center reaction imprint by the detection of a genome-wide mutational signature associated to the activity of AID (non-canonical AID or signature 9 [SBS9])[18,21,27]. The number of mutations associated to SBS9 significantly correlated with the IGHV gene identity observed both by SSeq and IgCaller (Supplementary Fig. 5). Therefore, SBS9 could help to corroborate the mutational status observed by IgCaller. However, the use of SBS9 alone would have miss-classified the Ig mutational status of 8 (5%) C1-CLL patients, highlighting that a proper analysis of the Ig gene rearrangement might be needed to correctly stratify patients based on the clinically-accepted cut off of 98% of identity (Fig. 2c).

**Ig light chain rearrangements**. A productive IGK or IGL gene rearrangement was found in 147 (97%) C1-CLL, 76 (97%) C2-CLL, 64 (100%) MCL, 27 (90%) MM, 45 (62%) DLBCL, and 7 (100%) B-NHL (Supplementary Data 9–14). These results were fully concordant with the IGLC expression observed by flow cytometry (FC) (Fig. 3a). Besides, we verified 5 randomly selected inversion-IGK productive rearrangements by SSeq (Fig. 1b, Supplementary Fig. 6, Supplementary Data 15). Furthermore,

IgCaller is also able to characterize the deletions occurring within the kappa deleting element (Kde) and the intron recombination signal sequence (RSS) allowing for a full characterization of the IGK locus[28]. In this regard, IgCaller identified 178 Kde-RSS deletions, 139 Kde-IGKV deletions, and 5 RSS-IGKV deletions (Supplementary Data 16). We confirmed the presence of these deletions by PCR in three selected cases (Supplementary Fig. 7). IgCaller also identified 246 unproductive/unexpressed IGK/L rearrangements in 177 cases (Supplementary Data 16). Considering that virtually all these cases expressed a productive IGK/L rearrangement, this finding emphasizes the sensitivity of IgCaller to detect multiple rearrangements.

The ability to determine the IGLC rearrangements from WGS data is of clinical relevance due to its prognostic value. In fact, we identified IGLV3-21 in 25/223 (11%) CLL, which was associated with a shorter time to first treatment (TTFT) in M-IGHV cases, and with a shorter overall survival independently of the IGHV mutational status, as recently suggested (Fig. 3b)[8].

**Class switch recombination**. IgCaller identified the CSR matching the isotype expressed by FC analysis in 27/30 (90%) MM (Fig. 3c, Supplementary Data 17). Two of the three potentially discordant cases with IgG or IgM by WGS expressed only IGLC by FC. In the latter case, two IGH translocations also detected by IgCaller caused the loss of the constant IGH region of both alleles leading to sole IGLC expression (Supplementary Fig. 8). Overall, IgCaller could not identify the isotype switch in 1 (3%) MM expressing IgG (Supplementary Fig. 8). Next, CSR was observed in 37/230 (16%) CLL, 2 (3%) MCL, 28 (38%) DLBCL [18, 55%, germinal center B cell subtype (GCB); 7, 23%, activated

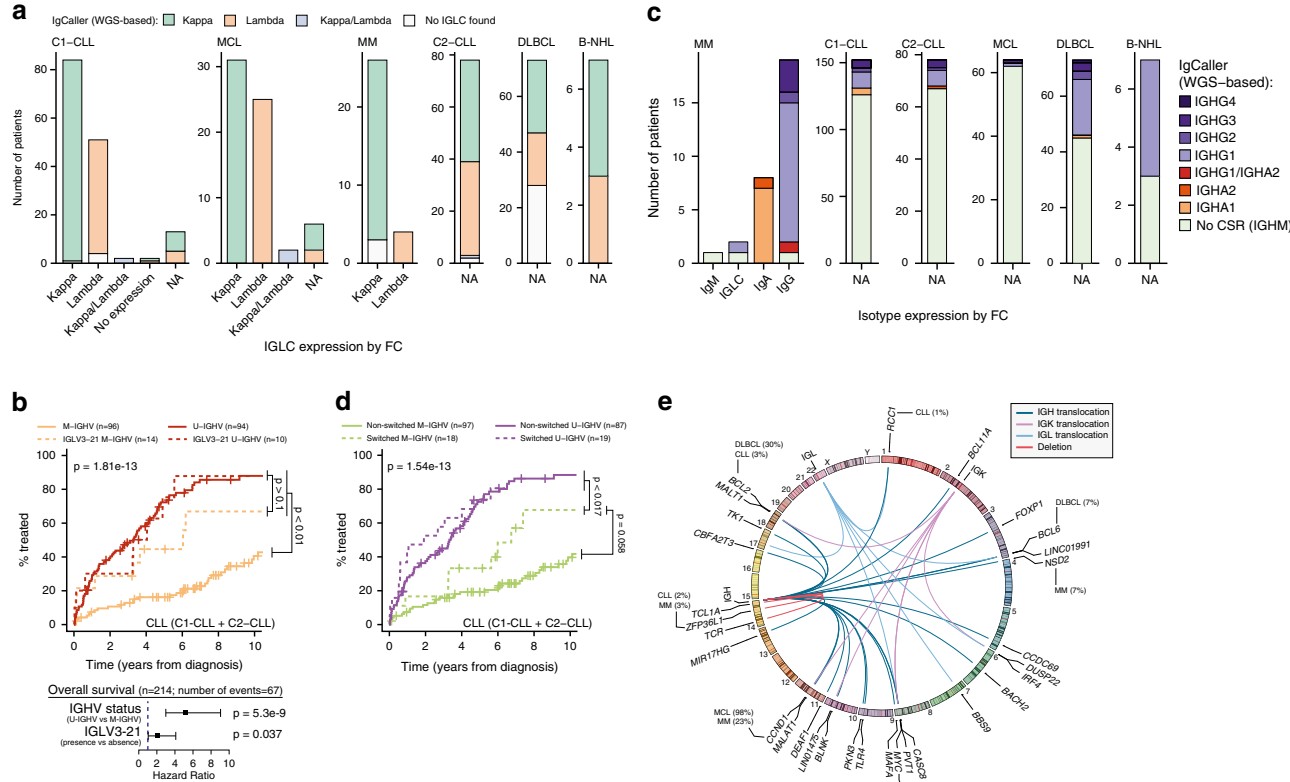

**Fig. 3 Benchmarking of IgCaller: IGLC rearrangements, CSR, and oncogenic Ig translocations. a** Agreement between the IGLC productive rearrangement detected by IgCaller and FC analysis. **b** TTFT and OS of patients with CLL according to the presence of IGLV3-21 rearrangements. *P* values for TTFT curves are from Gray test. *P* values for the multivariate analysis of OS are from Cox regression. **c** Comparison of the CSR identified by IgCaller and FC. **d** TTFT of CLL patients according to the presence of CSR. *P* values are from Gray test. **e** Circular representation of the oncogenic Ig rearrangements (translocations and deletions) identified by IgCaller genome-wide. Frequencies of recurrent alterations are shown. Source data are provided as a Source data file.

B cell subtype; and 3, 33%, unclassified], and 4 (57%) B-NHL (Fig. 3c, Supplementary Data 17–22). The distribution of CSR in the different tumor types is similar to that observed using FC or SSeq[9,29–31]. We confirmed the WGS-derived CSR in 6 randomly selected CLL cases by FC (Supplementary Fig. 9, Supplementary Data 18). Noteworthy, the presence of CSR in 18/115 (16%) M-IGHV identified CLL patients with a tendency to a shorter TTFT than non-switched M-IGHV CLL ($p = 0.058$, Fig. 3d). It is known that CLL with stereotypes #4 and #16, although expressing IgG, follow an indolent clinical course[9,32,33]. The stereotypy analysis of our CLL cases showed that none of them carried these specific subsets (Supplementary Data 2 and 3).

**Oncogenic Ig translocations**. IgCaller identified Ig translocations in 11 (7%) C1-CLL, 5 (6%) C2-CLL, 63 (98%) MCL, 15 (50%) MM, 34 (47%) DLBCL, and 2 (29%) B-NHL (Supplementary Data 23). FISH/PCR data available for 54 cases confirmed all the rearrangements identified by IgCaller in these tumors (Supplementary Data 23). As previously described, the most common Ig translocation in CLL was the t(14;18) [IGH-BCL2] in seven cases; the t(11;14) [CCND1-IGH, $n = 62$] or t(2;11) [IGK-CCND1, $n = 1$] in all but one MCL (note that this later case had the characteristics of a CyclinD1-negative MCL);[34] the t(11;14) [CCND1-IGH, $n = 7$], MYC-IG ($n = 4$), and t(4;14) [NSD2-IGH, $n = 2$] in MM; and the t(14;18) [IGH-BCL2, $n = 22$] and t(3;14) [BCL6-IGH, $n = 5$] in DLBCL (Fig. 3e). Of note, IgCaller identified three Ig rearrangements in two CLL cases [t(6;22) [IRF4-IGL] and t(1;22) [RCC1-IGL]; IGK insertion in LINC01475] that were not detected in previous WGS analyses (Supplementary Figs. 10 and 11)[18]. Similarly, three Ig translocations identified by IgCaller in DLBCL, two of them involving BCL6, were not previously reported (Supplementary Data 23)[22]. Altogether, these results emphasizes the sensitivity of IgCaller to detect oncogenic Ig translocation. Besides, the identification of chromosomal alterations within the Ig loci of these samples suggests that the ability of IgCaller to reconstruct the V(D)J rearrangement is not influenced by secondary Ig structural events.

**Effect of sequencing depth and tumor purity**. Two main factors that might influence the performance of IgCaller are the sequencing depth (or coverage) and the tumor cell content (or purity) of the sample. We did not observe a remarkable effect of the mean coverage of the Ig loci and the percentage of Ig productive rearrangements identified in the CLL and MCL cohorts (mean depth ranging from 11× to 96×) (Fig. 4a, Supplementary Data 2–4). To further analyze the effect of coverage, we next focus on 29 cases (21 C1-CLL and 8 MCL) with a mean depth >30×, tumor purity >90%, and carrying an identified IGH and IGK/L productive rearrangements by IgCaller. We randomly down-sampled the initial BAM files to mean coverages ranging from 5× to 30× (Methods). The sensitivity of IgCaller to detect the complete productive rearrangements was >0.9 at 20× and >0.85 at 15× (Fig. 4b, Supplementary Data 24). Sensitivity started to drop at 10× (0.72 for IGH and 0.79 for IGK/L), while <15% and <50% of the IGH and IGK/L rearrangements could be identified at 5×, respectively. Besides, two cases included in this analysis carried a CSR expressing IGHG1 that could be identified at 15×. Similarly, eight cases carried oncogenic translocations that were called at a minimum coverage of 10× (1), 15× (2), 20× (4), and 25× (1). Also of interest, the IGHV gene identity reported by IgCaller was minimally effected by sequencing depth (Fig. 4c). Altogether, these results suggest that IgCaller is robust at sequencing depths ranging from >15× to 100×, while some rearrangements can still be identified at lower coverage. We are

not aware of any potential limitation regarding the use of IgCaller with WGS data with >100× of depth.

To analyze the effect of the tumor cell content, we created in silico tumor samples at distinct tumor purities (ranging from 5% to 95%) by mixing at different ratios the previous 29 tumor samples with their respective non-tumoral WGS reads (Methods). At a final mean depth of 30×, the sensitivity of IgCaller to detect a complete IGH rearrangement was >0.85 with tumor purities >50%, while it was >0.8 with purity >20% for IGK/L (Fig. 4d, Supplementary Data 25). The limitation to detect a complete IGH rearrangement at tumor cell contents around 20–35% could be overcome at 60× of sequencing depth (Fig. 4e, Supplementary Data 26). Of note, the identity of the IGHV gene was minimally affected at tumor cell contents >35% when considering 30× WGS data (Fig. 4f). As expected, the accuracy of the IGHV gene identity was higher at 60× WGS, particularly for those samples with low tumor burden (Supplementary Data 26). The sensitivity of IgCaller was similarly influenced by an increasing contamination of a polyclonal-like population prepared by mixing 294 tumor samples (Fig. 4g, "Methods", Supplementary Fig. 12, Supplementary Data 27). As expected, in the context of polyclonal contamination IgCaller also identified Ig rearrangements present in the polyclonal population. Next, to demonstrate that IgCaller might be used to characterize oligoclonal samples, as suggested by the identification of multiple productive and/or unproductive IGH and IGK/L rearrangements, we mixed at different ratios two tumor samples carrying both a productive and an unproductive rearrangement caused by the presence of stop codon mutations ("Methods"). IgCaller was able to identify all four IGH rearrangements with scores calculated based on their number of reads that fitted with the pre-defined abundance of each tumor in each mixed sample (Fig. 4h, Supplementary Data 28). Overall, these data suggest that IgCaller is relatively stable when normal or polyclonal contamination is present within a clonal tumor sample. We have also shown that IgCaller is able to characterize oligoclonal scenarios carrying both productive and unproductive gene rearrangements.

**Comparison of IgCaller to other available algorithms**. Finally, we aimed to compare the performance of IgCaller to other available algorithms. After an exhaustive search we did not find any study that demonstrated that Ig gene rearrangements could be extracted from short-read WGS data. The lack of algorithms to reconstruct the Ig gene from short-read WGS data contrasts with the available set of programs to perform this analysis from repertoire sequencing (Rep-Seq) data, which specifically amplify the entire Ig sequence using long reads and specific primers[35] as well as RNA-seq data[36–38]. Nonetheless, one of the programs (MiXCR)[36] uses a general framework that allows the analysis of the B cell receptor from both RNA and DNA sequencing data by starting from raw sequences (FASTQ files). In order to compare the performance of both programs, we run MiXCR on 194 tumor samples (136 C1-CLL and 58 MCL). MiXCR identified the same IGH CDR3 sequence compared to SSeq/IgCaller in all cases analyzed, with only subtle differences in seven cases (Supplementary Data 29). Similarly, the same V, D, and J genes were identified in 174/194 (90%) cases. The remaining cases differed in the D or V gene (8%, 4%) or had multiple potential V or J genes reported according to MiXCR (12%, 6%). These discrepancies in the identification of the V, D and J genes are in line with the fact that MiXCR did not report the complete V(D)J sequence in any of the samples analyzed. The lack of the full V(D)J sequence impaired the characterization of the IGHV identity using MiXCR in the studied low-coverage, short-read WGS data (Supplementary Data 29).

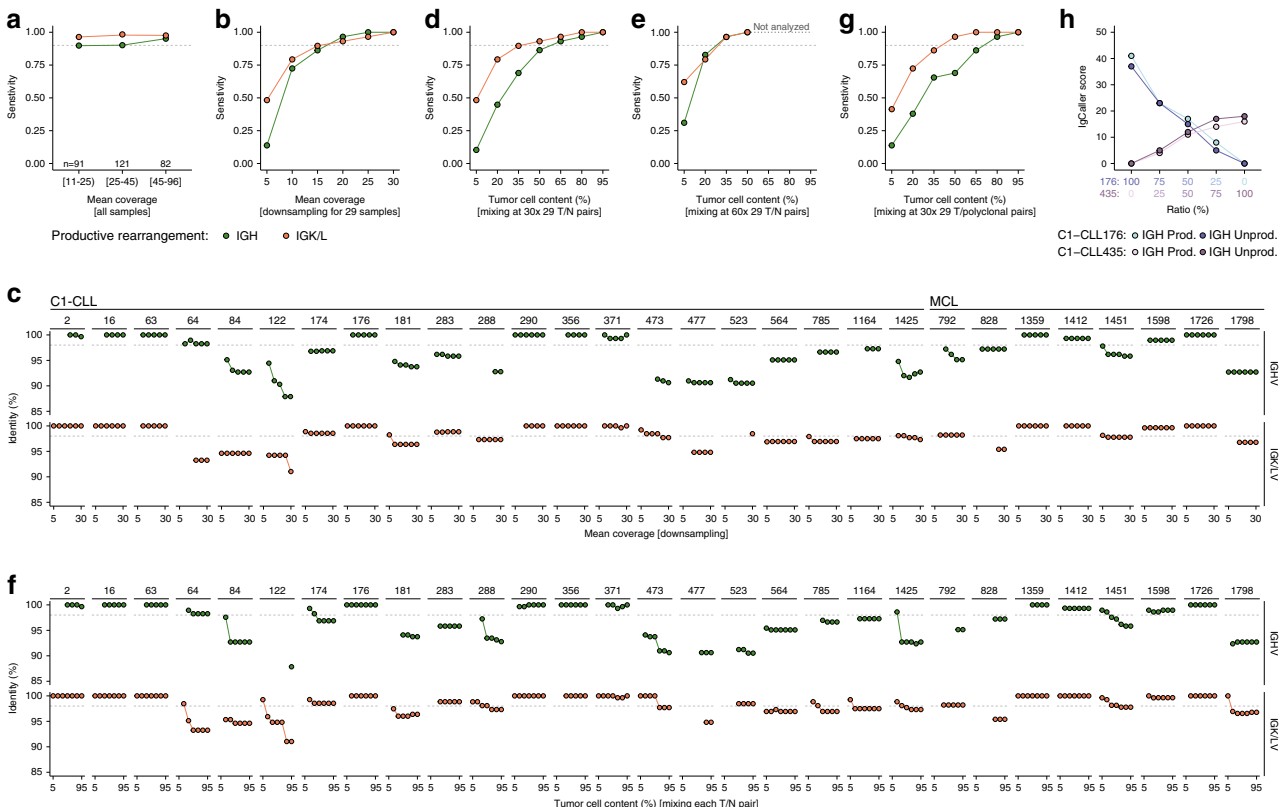

**Fig. 4 Sequencing depth and tumor purity requirements for IgCaller. a** Sensitivity of IgCaller to detect a complete and productive IGH or IGK/L rearrangement at different ranges of coverage for CLL and MCL cases. **b** Downsampling experiment with 29 tumor samples. The sensitivity of IgCaller is shown for each specific mean coverage analyzed. **c** Identity for IGH (top) and IGK/L (bottom) gene rearrangements for each case at different downsampling conditions. **d** Sensitivity of IgCaller at distinct tumor cell contents after mixing tumor/normal (T/N) pairs at different ratios. The mean depth was set to 30×. **e** Similar to **d** but with a mean depth of 60×. Note that only purities of 50%, 35%, 20%, and 5% were analyzed. **f** Ig gene identity according to tumor cell content in different T/N mixing conditions. **g** Sensitivity of IgCaller when tumor samples are mixed with a polyclonal-like population. **h** Oligoclonal situation created in silico by mixing at different ratios two tumor samples carrying two IGH rearrangements each; one productive (Prod.) and one unproductive (Unprod.). The score of each rearrangement according to IgCaller is shown. A score of 0 is used for illustrative purposes for rearrangements not identified in a specific mixing condition. The score is calculated based on the number of reads supporting each rearrangement ("Methods"). Source data are provided as a Source data file.

## Discussion

The characterization of Ig gene rearrangements and oncogenic translocations is an important diagnostic and prognostic parameter in different B cell neoplasms, and guide the management of the patients[2–4,6,10]. In spite of the expansion of WGS analyses in research and clinical settings, these rearrangements are still studied using SSeq, NGS, and/or FISH due to the inherent complexity of the Ig loci. In this report, we describe that the rearranged Ig genes of B cell neoplasms, including CSR and oncogenic Ig translocations, can be fully reconstructed from short-read WGS data. To this aim, we developed IgCaller, an algorithm that uses standard aligned BAM files to characterize the Ig gene rearrangements without the need of any additional preprocessing step. The usage of already aligned WGS data might facilitate the implementation of IgCaller in virtually any bioinformatic pipeline, and will contribute to elucidate the genomic landscape of B cell lymphomas and leukemias using a single approach[17].

IgCaller reconstructed the complete Ig gene (IGH and IGK/L productive rearrangements) of 79% of 404 B cell neoplams with >98% accuracy when compared with standard SSeq/NGS and FC analyses. The characterization of the complete Ig gene was higher in CLL (87%) and MCL (98%) than in MM (63%) and DLBCL (41%), probably due to the higher number of somatic mutations.

At least one IGH or IGK/L productive rearrangement was seen in 99% CLL, 100% MCL, 97% MM, and 81% DLBCL. The sensitivity of IgCaller is similar to that observed with SSeq[39]. Besides, we observed a highly significant correlation and concordance between the identity of the rearranged IGHV gene sequences obtained by IgCaller and SSeq/NGS. IgCaller also determined the presence of IGLC rearrangements, CSR and oncogenic Ig translocations of clinical value in these neoplasms. Of note, some Ig translocations detected by IgCaller were not recognized in previous analyses, emphasizing the sensitivity of our algorithm[18,22]. We have shown that IgCaller is stable at low sequencing depths (i.e. 10×), although its sensitivity increases with coverage. Similarly, normal in tumor contamination had a minimal effect on the sensitivity and specificity of IgCaller, especially when analyzing 60× WGS data. It is important to highlight that IgCaller is not designed to work with polyclonal samples (i.e. normal B cell populations); an analysis that indeed is impaired by the low coverage of WGS. To this aim, other available methodologies (Rep-Seq or RNA-seq)[35] and tools (such as MiXCR)[36] might be more appropriate. These approaches might allow also the analysis of B cell clonal evolution and/or ongoing somatic hypermutation. Nonetheless, we have shown that IgCaller is also able to characterize clonal tumor rearrangements in the context of contamination of a polyclonal-like B cell population. Furthermore,

IgCaller was able to identify multiple productive and unproductive Ig rearrangements within the same tumor sample allowing the characterization of oligoclonal tumor populations.

Altogether, the complete characterization of the rearranged Ig gene based on short-read WGS data, when available, could facilitate the analysis of IGLC rearrangements, CSR, and oncogenic Ig translocations, and replace the standard SSeq/NGS/FISH of the Ig loci both in research and clinical settings.

## Methods

**Input files**. IgCaller extracts the reads of interest from already aligned WGS data (BAM files) avoiding to re-align the entire data set with custom or specific tools. The functionality of this program was verified aligning the raw reads using the BWA-MEM algorithm (v0.7.15 and v0.7.17)[23] with default parameters, and converting SAM files to BAM files using either samtools (version 1.6 and 1.9)[24] or biobambam2 (v2.0.65, https://gitlab.com/german.tischler/biobambam2).These programs are widely accepted and virtually the default algorithms used in most cancer genomics projects[25]. However, IgCaller should work well (or be easily adapted to work well) with any BAM file obtained using any of the available algorithms designed to align and process paired-end WGS data.

In addition to the WGS BAM file of the B cell neoplasm of interest (hereafter tumor sample or tumor BAM file), the reference genome used to align the raw reads and/or the BAM file of the germ line of the patient (hereafter BAM file of the normal sample) are required to reconstruct the mutated rearranged sequence and to assess its identity to the germ line. If the normal BAM file is available, it is used to account for individual polymorphisms. If a given position is not fully covered in the normal BAM file, the nucleotide present in the reference genome is considered. If the reference genome is not available, these uncovered positions are not considered in the identity calculation and reported as N in the output sequence. If the normal BAM file is not available, the germ line sequence is directly extracted from the reference genome. Note that the reference genome does not include information regarding the presence of polymorphisms. Then, if the normal BAM file is missing, polymorphisms will be considered to be mutations rather than germ line polymorphisms, which will negatively influence the identity of the rearranged IGHV sequence detected by IgCaller. Therefore, in this scenario, we strongly recommend analyzing the rearranged sequence obtained by IgCaller using IMGT/V-QUEST or IgBLAST, which will account for polymorphisms as traditionally applied for tumor-only SSeq/NGS sequences.

The third required piece of information is the BED files containing the genomic locations of the V and J genes of IGH/IGK/IGL (wgEncodeGencodeBasicV19 for hg19, and GencodeV29 for hg38), the CSR regions (Huebschmann et al., in preparation)[26], and the sequences of the annotated D genes (extracted using the coordinates of the D genes reported in wgEncodeGencodeBasicV19 for hg19 and GencodeV29 for hg38). Although the user could define their own regions and sequences, these files are supplied within the IgCaller program both for hg19 and hg38. A set of optional parameters, which are described in a next section, might be specified when running IgCaller.

**Identification of V-J rearranged pairs and break points**. The first step of IgCaller consists of extracting the reads aligning to the Ig loci from the supplied tumor and normal BAM files using samtools, and a mini BAM file is temporarily created to speed up downstream executions. Reads that are not primary alignments, supplementary alignments, and PCR or optical duplicates are removed (-F 3318 option in samtools view). Once proper reads aligning to the regions of interest are extracted, two types of reads are considered to identify potentially rearranged V–J pairs: (i) split reads: soft clipped reads spanning the boundaries of a V and J gene. Split reads only mapping to a V or J gene but with >20 bp soft clipped bases (S in CIGAR) with no mapping information (no "SA:" field in the aligned read) are also labelled as split reads and used as explained below. (ii) Abnormal insert size reads: read pairs with anomalous insert size (here defined as insert size >10,000 bp) in which one read maps to a V gene and its pair aligns to a J gene. Note that one pair of reads might be considered as both split and abnormal insert size reads due to the fact that one read might be a split read spanning the V–J boundaries while its pair might map to the J or V gene. It is important to notice here that reads do not map to the D gene of the IGH locus due to is small length. Once split reads and abnormal insert size reads fulfilling the previous considerations are identified, the specific V and J break points are identified based on the split reads spanning both genes. Then, a score based on the number of split reads (2 points for each read) and abnormal insert size reads (1 point for each pair) is given to each V–J pair to discriminate likely real V–J rearranged pairs from random sequencing artifacts. Note that if more than one pair of break points are found for a given V–J, all of them are kept and considered downstream, but each specific break will have a different score based on its number of split reads. Besides, if the tumor purity is known, these scores are adjusted by the tumor cell content of the BAM file to increase the sensitivity in samples with low tumor purity as well as to make comparable the rearrangements/scores obtained for different samples.

In the scenario that a given V–J pair is not supported by split reads spanning both genes (i.e. only identified by abnormal insert size reads), the precise break

points cannot not be identified. This is more likely to occur with extremely short-read sequencing experiments (i.e. $2 \times 90$ bp, as some of the samples within the C1-CLL cohort) than with the new available read lengths (i.e. $2 \times 125$ bp or $2 \times 150$ bp, as applied in C2-CLL, MCL, and MM cohorts). However, if this situation occurred, all potential combination of breaks coming from split reads mapping only to the V and to the J genes would be considered.

Once all potential V–J pairs are identified, each of them with their specific break points, V and J sequences are extracted using samtools mpileup from the start of the gene to the break point (or from the break point to the end of the gene, depending on the gene and its orientation in the human genome, see Fig. 1b). If available, the germ line sequence is obtained from the normal BAM file using the same set of coordinates/break points, and individual polymorphisms are considered when assessing the percentage of mutations found in the tumor sample.

In order to increase the sensitivity of our approach, once the J and V sequences are obtained, split reads mapping only a J or V break of one of the previously identified J–V pairs with a soft clipped unmapped sequence of >20 bp are considered to span a V–J pair if ≥5 bp of the soft clipped unmapped sequence can be mapped to the second break. Thanks to this step, reads spanning J–V boundaries can be rescued to better discriminate true from artifactual rearrangements. Besides, it may allow for the identification of the specific break points of a given V–J pair previously identified by sole abnormal insert size reads.

**Extraction of the unmapped junction**. Once IgCaller has identified the V–J breaks, the N nucleotides-D gene-N nucleotides sequence (N-D-N, for IGH) or N nucleotides (N, for IGK and IGL) between V and J genes are obtained from the unmapped fraction of the split reads spanning the boundaries of both genes. Thus, the unmapped sequence of all split reads spanning each specific V-J break are retrieved, those with the most common length are kept, and only the most abundant nucleotide in each position is considered to report a unique consensus N-D-N/N sequence for each V–J specific pair. Note here that if the two most common N-D-N/N lengths have the same number of supporting reads IgCaller reports both potential rearrangements. Similarly, if all potential N-D-N/N sequences have different lengths, all of them are reported. To obtain the most likely D gene, a Smith-Waterman alignment of the N-D-N sequence obtained is performed against all D gene sequences supplied with a match score of 5 and mismatch and gap costs of 4 and 8, respectively. We noticed some discrepancies in around 10% of the cases analyzed regarding the D gene obtained from IgCaller's Smith-Waterman in comparison to the D gene reported by IMGT/V-QUEST likely due to differences in the alignment procedure. Therefore, we recommend using IMGT/V-QUEST or IgBLAST to confirm the D gene reported by IgCaller.

**Functionality and identity of the reconstructed sequence**. Although the aim of IgCaller is to retrieve the rearranged sequences of the Ig gene from short-read WGS rather than to completely assess their functionality (to this aim there are specific tools such as IMGT/V-QUEST and IgBLAST), IgCaller performs an assessment of the functionality of the V(D)J (IGH) and VJ (IGK/IGL) sequences obtained. This assessment is performed by identifying the cysteine (C) 23, tryptophan (W) 41, cysteine 104, and phenylalanine (F) or tryptophan 118 at the conserved FGXG or WGXG motif (G, glycine; X, any amino acid). Then the CDR3 region (flanked by C104 and F/W118) is translated to assess its productivity (i.e. in-frame or out-of-frame junction). Note that IgCaller removes short insertions within the V gene before assessing the functionality of a rearranged sequence. In this scenario, a tag is added in the predicted functionality specifying that indels were found. That said, note that short insertions are retained in the output sequence of IgCaller to retrieve the original sequence as would be obtained by SSeq/NGS. Besides, IgCaller also searches for stop codons within the FR1-FR3 sequence. Although IgCaller was able to identify the correct functionality of around 95% of the obtained sequences when compared to IMGT/V-QUEST, we suggest to use specific tools to corroborate these results.

To assess the identity of the rearranged sequence, the germ line sequence is obtained from the normal BAM file (if available) or the reference human genome. The percentage of identity is calculated within the identified FR1 to FR3 regions. If IgCaller fails in identifying the last C104, the entire sequence of the V gene is used to calculate an approximate identity. Both the percentage of identity and the number of nucleotides considered in this calculation are reported.

**Locus specific considerations**. Due to the inherent differences in the processing and/or orientation of IGH, IGK, IGL and CSR, each of the analysis performed by IgCaller have some peculiarities that need to be taken into account for a proper analysis.

The IGK locus, in addition to the V and J genes, also includes the so-called intronic recombination signal sequence (RSS, downstream of the J genes) and the kappa deleting element (Kde, found 24 kb downstream of the constant K region). Once an unproductive IGK rearrangement is formed, a deletion within the Kde and RSS may occur to eliminate the constant and enhancer region of the IGK preventing the expression of the unproductive rearrangement. Similarly, a deletion involving the Kde and a given V gene may occur to completely eliminate the unproductive rearrangement[28]. Both Kde-RSS and Kde-V gene deletions are investigated by IgCaller to further characterize this locus. Moreover, although the

IGKJ genes and proximal IGKV genes are oriented on the negative strand, the IGKV4-1, IGKV5-2, and IGKV genes within the distal cluster are inverted and therefore oriented on the positive strand. Thus, rearrangements involving the latter IGKV genes occur by inversions of the IGKV genes rather than deletions.

Regarding the IGL locus it is important to notice that all V and J genes are oriented in the forward orientation relative to the genome build while IGH is oriented in the reverse orientation.

For class or isotype switching, IgCaller searches for deletions within the described CSR of IGHM and IGHA1/2, IGHE or IGHG1/2/3/4 (Huebschmann et al., in preparation)[26]. After an inspection of the CSR regions we observed that split reads may confound a proper identification of the deletions within the CSR due to their repetitive nature, high similarities within CSR regions, and presence of a remarkable number of variants and/or sequencing artifacts. Therefore, the use of split reads in CSR analysis hindered a robust identification of the exact break points and added noise to the general identification of the potential deletions. As the exact break points within the CSR are not required to properly assess the isotype switch, we decided to exclude split reads in this specific analysis. However, in addition to determine a potential deletion by abnormal insert size reads, the coverage of a 1500 bp window upstream and 1500 bp window downstream of the CSR identified is compared to assess for a drop of coverage within the deleted region, which will emphasize the presence of the CSR. For example, if a deletion occurs within the IGHM and IGHG1, a significant reduction of coverage (or read depth) should be observed in the region within the IGHM-IGHG1. Therefore, following the previous example, the coverage of a 1500 bp window upstream of the CSR of the IGHG1 and a 1,500 bp window downstream of the CSR of IGHG1 is compared using a Wilcoxon test. Potential class switch deletions with no reduction of coverage are automatically removed. If the normal WGS is available, the coverage at each position of the window in the normal BAM file is subtracted to that of the tumor sample to consider for fluctuations in coverage that are sequence/region specific. Besides, as applied in the calculation of the score based on the number of reads, the reduction of the coverage is adjusted by the tumor purity, if available, to increase the sensitivity of IgCaller in samples with low tumor content. This comparison of coverage enhanced the specificity of the CSR detection.

**Pre-defined filter of low evidence rearrangements**. IgCaller performs a pre-defined filter to highlight high confidence rearrangements while separating them from low evidence, likely artefactual, sequences. However, both high and low confidence rearrangements are stored in locus-specific output files. The main two filters applied to the IGH, IGK and IGL sequences are the requirement of a score ≥3 (after adjusted by the tumor cell content of the sample), and that more than half of the J and V sequences must contain a nucleotide other than N. Then, if two exact V(D)J/VJ rearrangements or highly similar V(D)J rearrangements (i.e. sharing two of the three genes) are found, IgCaller keeps the productive one with the highest score. For IGK, if IgCaller finds two rearrangements involving the same IGKJ gene and the same IGKV gene from the proximal and distal cluster, respectively (e.g. IGKJ2-IGKV2-40 and IGKJ2-IGKV2D-40), it keeps the rearrangement with the highest score. If both have the same score, the one involving an IGKV gene of the proximal cluster, which usually has the higher mapping quality, is kept.

When analyzing the CSR, rearrangements considered as high confidence must have a mean coverage >8× in the upstream window analyzed (the one not affected by the deletion), a score ≥4 coupled with a reduction of the mean coverage of the two windows ≥60% or a score ≥7 with a mean reduction of ≥30%, and a p value by Wilcoxon test <1e-10. This double combination of score and reduction of coverage allows CSR strongly supported either by a remarkable drop of read depth or by a high number of reads spanning the deletion. Note that if more than one class switch deletion passes the previous filters, only the one with the highest reduction of coverage will be reported in the high confidence output file. In our experience, nearly all IGH V(D)J sequences identified by SSeq/NGS, productive kappa or lambda rearrangements matching the light chain expression, and isotypes identified by FC were called as high confidence rearrangements within the high confidence passing rearrangements. However, we recommend to check low-confidence rearrangements specially when analyzing low coverage and/or very short-read (i.e. 90 bp) WGS data, or if there is an interest in identifying minor subclonal populations carrying distinct Ig rearrangements.

**Genome-wide analysis of oncogenic Ig rearrangements**. In addition to characterize the rearranged V(D)J sequences and the presence of CSR, IgCaller searches for Ig rearrangements genome-wide. In this regard, any potential deletion, inversion or gain with one break within any of the Ig loci (IGH, IGK or IGL) is annotated if more than $X$ (4 by default) abnormal insert sizes and/or split reads mapping to a distant location of the chromosome (>10,000 bp) are found within a distant <1000 bp from one another. Translocations are identified based on read pairs in which one read maps to an Ig locus while its pair maps to a different chromosome. Split reads mapping to both chromosomes are also considered. The break points reported correspond to the most 5′ position in the positive strand, and the most 3′ position in the negative strand. Considering that to speed up the execution of IgCaller it only works with reads aligned to the Ig loci, the non-Ig break point might not be called precisely at a single-base resolution if split reads supporting the rearrangement are not found. In this scenario, the approximate non-Ig break point is extracted from the starting alignment location of the non-Ig

reads (note that the exact break point would correspond to the ending alignment location for rearrangements/reads mapping to the positive strand, but this information could only be retrieved looking at the CIGAR information of that read).

Next, to exclude artifactual rearrangements, the normal BAM file is used, if available, to annotate the number of reads supporting each potential rearrangement in the normal sample. In this scenario, a ± 1000 bp window is considered from the region in which reads supporting the rearrangement were observed in the tumor sample. Finally, using the default parameters, rearrangements supported by a score ≥10 in the tumor sample and ≤2 reads in the normal BAM file are considered as high confidence and reported within the passing rearrangements. Besides, both high and low confidence rearrangements are reported in a specific file for the genome-wide oncogenic Ig rearrangements. Due to the well-known difficulties on the detection of rearrangements genome-wide, we recommend to manually review the alterations found using a visualization tool such as the Integrative Genomics Viewer (https://software.broadinstitute.org/software/igv) to filter out potential artifactual rearrangements that could have passed the permissive filters of IgCaller.

**Running IgCaller**. To run IgCaller it is necessary to have python3 installed with the following modules: subprocess, sys, os, itertools, operator, collections, statistics, argparse (v1.1), regex (v2.5.29 and v2.5.30), numpy (1.16.2 and v1.16.3), and scipy (v1.2.1 and v1.3.0). Although providing the versions of the previous modules tested, we are not aware about any specific version requirement for running IgCaller. The only required non-python program is samtools (version 1.6 and 1.9 have been tested).

When running IgCaller the user must define a few mandatory arguments:

inputsFolder (-I): path to the folder containing the supplied IgCaller reference files.
genomeVersion (-V): version of the reference human genome used when aligning the WGS data (hg19 or hg38).
chromosomeAnnotation (-C): chromosome annotation [ensembl = without "chr" (i.e. 1); ucsc = with 'chr' (i.e. chr1)].
bamT (-T): path to tumor BAM file.
bamN (-N): path to normal BAM file, if available.
refGenome (-R): path to reference genome FASTA file (not mandatory, but recommended, when specifying a normal BAM file. Mandatory when bamN not specified).

There are also some optional arguments:
pathToSamtools (-ptsam): path to the directory where samtools is installed. There is no need to specify it if samtools is found in PATH (default = empty, assuming it is in PATH).
outputPath (-o): path to the directory where the output should be stored. Inside the defined directory IgCaller will automatically create a folder named *tumorSample*_IgCaller where output files will be saved (default, current working directory).
mappingQuality (-mq): mapping quality cut off to filter out reads for Ig V(D)J reconstruction (default = 0).
baseQuality (-bq): base quality cut off to consider a position in samtools mpileup when reconstructing both normal and tumor sequences (default = 13).
minDepth (-d): depth cut off to consider a position (default = 1).
minAltDepth (-ad): alternate depth cut off to consider a potential alternate nucleotide (default = 1)
vafCutoffNormal (-vafN): minimum variant allele frequency (VAF) to consider a nucleotide when reconstructing the germ line sequence using the supplied normal BAM file (if available) (default = 0.2).
vafCutoff (-vaf): minimum VAF to consider a nucleotide when reconstructing the tumor sequence (default = 0.1). Try to increase this value if only unproductive rearrangements are found due to stop codons. We have observed that relatively high coverage WGS (i.e. 100×) might carry many variants (likely sequencing artifacts) at VAFs around 10–20%.
tumorPurity (-p): purity of the tumor sample (i.e. tumor cell content) (default = 1). It is used to adjust the VAF of the mutations found in the tumor BAM file before filtering them using the vafCutoff, to adjust the score of each rearrangement, and to adjust the reduction of read depth in the CSR analysis.
minNumberReadsTumorOncoIg (-mntonco): minimum score supporting an Ig rearrangement in order to be annotated (default = 4).
minNumberReadsTumorOncoIgPass (-mntoncoPass): minimum score supporting an Ig rearrangement in the tumor sample in order to be considered as high confidence (default = 10).
maxNumberReadsNormalOncoIg (-mnnonco): maximum number of reads supporting an Ig rearrangement in the normal sample in order to be considered as high confidence (default = 2).
mappingQualityOncoIg (-mqOnco): mapping quality cut off to filter out reads when analyzing oncogenic Ig rearrangements (default = 15).
numThreads (-@): maximum number of threads to be used by samtools (default = 1).
keepMiniIgBams (-kmb): should IgCaller keep (i.e. no remove) mini Ig BAM files used in the analysis? (default = no).

As example, the command line to execute IgCaller would be:

python3 path/to/IgCaller/IgCaller_v1.py -I path/to/IgCaller/IgCaller_reference_files/ -V hg19 -C ensembl -T path/to/bams/tumor.bam -N path/to/bams/normal.bam -R path/to/reference/genome_hg19.fa -o path/to/IgCaller/outputs/.

IgCaller was tested on a MacBook Pro (macOS Mojave), Ubuntu (16.04 and 18.04), and MareNostrum 4 (Barcelona Supercomputing Center, SUSE Linux Enterpirse Server 12 SP2 with python3/3.6.1). IgCaller only requires 1 CPU, and it usually takes <2–5 minutes to characterize the complete Ig gene of one tumor sample. A longer execution time might reflect the identification of an unusual larger number of potential rearrangements. A demo data set to run IgCaller is provided along with the algorithm.

**Outputs of IgCaller**. The files generated by IgCaller are stored in the output directory (if specified) or in the current working director inside a folder called *tumor_sample*_IgCaller, where *tumor_sample* is the name of the supplied tumor BAM file (i.e. *tumor_sample*.bam). Inside this folder, IgCaller stores several temporary files, which are removed once the execution finishes, and the following final output files:

tumor_sample_output_filtered.tsv: High confidence rearrangements passing the pre-defined filters are stored in this file (an example along with a description of the different fields might be found in Supplementary Data 30).
tumor_sample_output_IGH.tsv: file containing all IGH rearrangements identified by IgCaller (Supplementary Data 31).
tumor_sample_output_IGK.tsv: file containing all IGK rearrangements identified by IgCaller (Supplementary Data 31).
tumor_sample_output_IGL.tsv: file containing all IGL rearrangements identified by IgCaller (Supplementary Data 31).
tumor_sample_output_class_switch.tsv: file containing all CSR rearrangements identified by IgCaller (Supplementary Data 32).
tumor_sample_output_oncogenic_IG_rearrangements.tsv: file containing all oncogenic Ig rearrangements (translocations, deletions, inversions, and gains) identified genome-wide (Supplementary Data 33).

**Mutational signature analysis**. We used the previously published mutational data of the 152 C1-CLL cases[18,19]. Mutational signature analysis was performed as recently described[21]. Briefly, we de-novo extracted the mutational signatures found in the C1-CLL cohort using a non-negative matrix factorization. Signatures extracted were compared to the single base substitution (SBS) signatures reported in COSMIC (https://cancer.sanger.ac.uk/cosmic), and the one with the highest cosine similarity was kept. We identified the presence of signature 1 (or SBS1 in COSMIC), signature 5 (SBS5), signature 8 (SBS8) and signature 9 (SBS9). Next, we measured the contribution of each signature in each case using a fitting approach (MutationalPatterns R package). To avoid the inter-sample bleeding of signatures[21], we iteratively removed the least contributing signature in each case if the cosine similarity of the reconstructed mutational profile decreased <0.01. Due to their presence in all normal and tumor tissues, signatures 1 and 5 were always included if their addition increased the cosine similarity[27]. The presence/absence of SBS9 was used as a surrogate of the mutational status of the C1-CLL patients (M-IGHV or U-IGHV, respectively). An R script is available within IgCaller to facilitate this analysis.

**Orthogonal verification of Ig gene rearrangements**. The sequence analysis of the IGH V(D)J rearrangements by SSeq was performed on either genomic DNA or complementary DNA using leader or consensus primers for the IGHV FR1 along with appropriate consensus constant primers[18]. IGK rearrangements were amplified using previously described primers[12] on 20 ng of genomic DNA. PCR amplifications were performed using the Taq PCR MasterMix Kit (Qiagen), and run on a QIAxcel Advanced System (Qiagen). Sanger sequencing was performed on an ABI Prism BigDye terminator (Applied Biosystems).

The LymphoTrack IGHV Leader Somatic Hypermutation Assay (Invivoscribe Technologies) was used to characterize IGH V(D)J rearrangements in 68 cases from the C2-CLL cohort. Libraries were performed using 50 ng of genomic DNA according to manufacturer recommendations. The bioinformatic analysis was done using the LymphoTrack MiSeq Data Analysis (version 2.3.1), similar to previous studies[11]. Briefly, we considered the top 10 reads after merging those reads that only differed from 1 or 2 nucleotides, and only reported rearrangements that accounted for >2.5% of the total number of reads per sample with >90% of the V region covered. These filters were used to remove rearrangements reflecting potential contamination of normal B cells and likely artefactual sequences, respectively.

Isotype and light chain expressions were assessed in the laboratory of their respective hospitals using the antibodies that were routinely tested in each specific time of assessment.

**IMGT/V-QUEST and ARResT/AssignSubsets**. The online IMGT/V-QUEST tool was run using default parameters searching for insertions and deletions in V-region[13,14]. The online ARResT/AssignSubsets tool was used to study stereotypy in CLL cases[16].

**Downsampling and polyclonal-like WGS data**. Downsampling of 29 tumor BAM files at specific mean sequencing depths (5×, 10×, 15×, 20×, and 25×) was performed using the samtools. The command used was: *samtools view -b -s frac input.bam > output.bam*, where *frac* was the fraction of reads from the initial BAM file to keep for each sample and downsampling condition. Similarly, to study normal in tumor contamination we first downsampled at 30× of mean coverage each tumor and normal pair. Then, we mixed at different proportion each tumor and normal pair. Tumor purities studied were 5%, 20%, 35%, 50%, 65%, and 80% at 30× of coverage. We also analyzed 5%, 20%, 35% and 50% of purity at 60×. The tumor cell content of the initial 29 tumor samples was summarized as 95% in these analyses considering that all of them had tumor purities >90%. To create an in silico polyclonal-like sample we merged 294 tumor BAM files (CLL and MCL) using samtools. Then we randomly downsampled this merged BAM file to 30×. We next mixed the 29 tumor samples used in the previous analyses with this polyclonal-like 30× WGS BAM file at final tumor cell contents of 5%, 20%, 35%, 50%, 65%, and 80%. A oligoclonal situation was created in silico by mixing at different proportions (0%, 25%, 50%, 75%, and 100%) two CLL samples both carrying a productive and unproductive IGH gene rearrangement. Note that we used a different seed in samtools for each experiment (downsampling, normal in tumor contamination, polyclonal-like contamination, and oligoclonal situation) to increase the randomness of the analyses. IgCaller was run using default parameters in all these experiments.

**MiXCR specifications**. We run MiXCR (version 3.0.12) using the following command:
*mixcr analyze shotgun -s hsa --starting-material dna --receptor-type bcr --contig-assembly sample_A_1.fastq.gz sample_A_2.fastq.gz output_folder*

MiXCR was run for C1-CLL and MCL samples with available raw sequences to avoid any bias in the preparation of FASTQ files from previously aligned WGS data. For each sample, the most abundant Ig clone carrying a productive IGH rearrangement was used for comparison. The output sequences (called targetSequences in the output of MiXCR) were used as input of the IMGT/V-QUEST online tool to compare the functionality and identity of the sequences obtained. We finally compare the V(D)J rearrangement and CDR3 sequence identified by MiXCR, SSeq, and IgCaller.

**Statistical analyses**. Comparison of the percentage of identity to the germ line between SSeq/NGS and IgCaller was performed using the Passing-Bablok regression (mcr R package v1.2.1), Pearson correlation coefficient (stats R package v3.5.0), and Bland-Altman plot (BlandAltmanLeh R package v0.3.1). The clinical relevance of specific Ig rearrangements was assessed for time to first treatment (TTFT) and overall survival (OS) calculated from the date of diagnosis to the date of first treatment/last follow-up and to the date of death/last follow-up, respectively. Disease-unrelated deaths were considered as competing events in TTFT analyses. Cumulative incidence curves of TTFT were compared using the Gray test. Multivariate analyses of OS were modeled using Cox regression. Clinical analyses were performed using the survival (v2.42-3) and cmprsk (v2.2-7) R packages. All tests were two-sided. Based on the analyses performed, we did not apply any adjustment for multiple comparisons. All analyses were performed in R (version 3.5.0).

**General considerations**. The study was approved by the Hospital Clínic of Barcelona Ethics Committee. Informed consent was obtained for all patients.

**Reporting summary**. Further information on research design is available in the Nature Research Reporting Summary linked to this article.

## Data availability
Sequencing data of the C1-CLL cohort, four MCL samples, MM, and DLBCL is available from the European Genome-phenome Archive (EGA) under accession numbers EGAS00001001306, EGAS00001000510, EGAS00001001299, and EGAS00001002936, respectively. Previously unpublished data has been deposited at EGA under accession numbers EGAS00001004165 (for tumor/normal WGS of 57 MCL cases) and EGAS00001004298 (Ig reads for C2-CLL and B-NHL cohorts as well as 3 MCL cases). All other data are included in the supplemental information or available from the authors upon reasonable requests. Source data are provided with this paper.

## Code availability
IgCaller is free-software and is available at https://github.com/ferrannadeu/IgCaller. Source data are provided with this paper.

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

## Acknowledgements

We are indebted to the Genomics Core Facility of the Institut d'Investigacions Biomèdiques August Pi i Sunyer (IDIBAPS) for the technical support, to R. Siebert and D. Huebschmann for sharing the CSR regions, and to K. Stamatopoulos, E. Vlachonikola and F. Psomopoulos for their helpful comments on the manuscript. We thank R. Eils, P. Lichter, C. von Kalle, S. Fröhling, H. Glimm, M. Zapatka, S. Wolf, K. Beck, and J. Kirchhof for infrastructure and pipeline development within DKFZ-HIPO and NCT POP. This study was supported by the Instituto de Salud Carlos III and the European Regional Development Fund "Una manera de hacer Europa" (PMP15/00007 to E.C.), the "la Caixa" Foundation (CLLEvolution-LCF/PR/HR17/52150017, Health Research 2017 Program HR17-00221 to E.C.), the National Institute of Health "Molecular Diagnosis, Prognosis, and Therapeutic Targets in Mantle Cell Lymphoma" (P01CA229100 to E.C.), and CERCA Programme/Generalitat de Catalunya. F.N. is supported by a pre-doctoral fellowship of the Ministerio de Economía y Competitividad (BES-2016-076372). F.M. is supported by the Memorial Sloan Kettering Cancer Center NCI Core Grant (P30 CA 008748). E.C. is an Academia Researcher of the "Institució Catalana de Recerca i Estudis Avançats" (ICREA) of the Generalitat de Catalunya. This work was partially developed at the Centre Esther Koplowitz (CEK, Barcelona, Spain).

## Author contributions

F.N. designed the study, collected and analyzed data, built and benchmarked IgCaller, prepared figures, and wrote the paper. R.M.V. analyzed data, built and benchmarked IgCaller, and wrote the paper. A.N., S.M., N.V., H.S.C., R.M., A.E., A.R.D., M.A., D.C., and S.B. performed wet lab experiments, collected data, and/or interpreted results. R.R., J. L., R.D.M., T.Z., X.S.P., and P.J.C. collected and/or analyzed data. J.D., T.B., and T.Z. collected clinical data. F.M. collected and analyzed data, and contributed to the conception of the study. E.C. designed the study, collected and analyzed data, and wrote the paper. All authors read, commented on, and approved the paper.

## Competing interests

The authors declare no competing interests.
