## [Peer Review File · Nature Communications]

Reviewers' comments:

Reviewer #1 (BCR specificity, system immunology (BCR biased))(Remarks to the Author):

Nadeu et al. present a data analysis approach for identifying the immunoglobulin gene sequences from whole genome sequence data of B cell-derived cancers. The method is generally very clearly explained, is freely available to other researchers, and is applied to a large set of published and unpublished genomes. This approach could be useful in the study of these cancers, particularly in cases where the Ig receptor may have a signaling role in pathogenesis. The current manuscript does not address the biological significance of the receptors identified, but the approach and software is likely to be of value to other researchers in the field.

Specific comments:

The authors do not address in much depth the effect that normal human variants in the Ig loci, such as copy number variants, will have on the IgCaller results. Were any of the alternative IGH loci sequence regions in the hg38 assembly tested in the analysis to see whether the results would be unaffected by alternative reference loci?

It would be important to determine the effect of the fraction of malignant clonal B cells present in a sample on the quality and accuracy of the IgCaller results. In samples that have significant numbers of non-malignant polyclonal B cells present in them, how is the IgCaller analysis affected?

The data disclosure should include the sequence reads from the currently unpublished cases whose Ig loci are used and analyzed in the study. To enable others to replicate the results, the reads from these genome regions should be deposited to a publicly-available repository (via dbGAP if required), rather than requiring readers to submit a request to the authors.

Reviewer #2 (B response, leukemia, cancer)(Remarks to the Author):

The manuscript „IgCaller: Reconstructing the Immunoglobulin Gene Rearrangements and Oncogenic Translocations in Lymphoid Neoplasms from Whole-Genome Sequencing” by Nadeu et al. reports an algorithm designated IgCaller which uses short-read WGS (whole-genome sequence) data to characterize immunoglobulin (Ig) gene rearrangements, sequence alterations and oncogenic translocations.

Generally, the VDJ gene rearrangement is central for determining the clonality of a B cell population in lymphoid neoplasms whereby individual VDJs serve as markers for the diagnosis of lymphoid neoplasms. Moreover, the mutational status of Ig genes is associated with specific clinical outcome in lymphoid malignancies such as chronic lymphocytic leukemia (CLL) or mantle cell lymphoma (MCL). Therefore, the identification of Ig gene mutations as compared with reference germ line sequence is an important step in the diagnosis of CLL for instance.

I found the paper interesting and I can appreciate the usefulness of the IgCaller for clinicians or scientists who study lymphoid malignancies and routinely use WGS. However, the manuscript describes a method and lacks clear novel finding.

For instance, what is the addition to the current knowledge about B cell malignancies?

Also, what is surprising is the conclusion that the presence of class switch recombination (CSR) in M-IGHV (16 of 106 cases or 15%) “identified CLL patients with a tendency to a shorter time to first treatment (TTFT) than non-switched M-IGHV CLL (Fig. 3d)”. However, stereotype subset#4 CLL is mutated, expresses IgG (thus switched) and shows a very indolent course of disease as it mostly does

not requires therapy. This is in sharp contrast to the authors' conclusion.

Reviewer #3 (B cell biology, BCR, VDJ rearrangement)(Remarks to the Author):

In this report, Nadeu and colleagues describe IgCaller, a versatile bioinformatic tool aimed to reconstruct the identity of clonal Immunoglobulin (Ig) heavy (H) and light (L) chain V gene rearrangements from short sequencing reads obtained from whole genome sequencing (WGS) of primary specimens of human B cell malignancies.

The Ig caller algorithm offers the opportunity to combine Ig VDJ profiling with the identification of the possible IgH isotype expressed by the clonal malignant B cell population, and to the mapping at nucleotide resolution of the breakpoint of possible chromosomal translocations involving IgH/L chain gene loci.

IgCaller is the latest of a series of algorithms developed in recent years to extrapolate Ig VH/VL gene repertoires from healthy and/or pathological B cell populations using Next Generation Sequencing (NGS)-based approaches. IgCaller complements current algorithms as it exploits short sequencing reads generated by WGS. Such practice is yet rarely employed to identify the identity of Ig V gene clonotypes defining pathological B cell population(s) contributing to disorders such as B cell neoplasms.

IgCaller exploits two major features to reconstruct unique V(D)J joining regions, namely so-called split reads and abnormal insert size reads, after annealing short reads coming from WGS to enriched genomic regions covering IgH and IgL chain loci. Such principle has been previously employed by other Ig repertoire reconstruction algorithms, which are centered on the analysis of NGS reads obtained from Ig V gene targeted and whole exome sequencing datasets. The present manuscript lacks any form of bench marking of IgCaller with currently available algorithms reconstructing IgV gene rearrangements from NGS short reads.

The present manuscript lacks information on the minimum WGS coverage required to faithfully reconstruct IgVH/VL gene clonotypes representative of malignant B cells within the tumor biopsy. It remains also undetermined the performance of the algorithm under conditions where putative clonal B cell population(s) in a starting tumor are overwhelmed by a majority of polyclonal wild-type B cells. Testing the performance of the algorithm under controlled conditions where WGS data are produced from a unique tumor population (i.e. a tumor cell line) mixed at different ratios with a polyclonal B cell population may provide useful information on the sensitivity of an approach that may suffer from a possible low coverage of the relevant genomic data.

The analyses, using IgCaller, of IgH/L V gene rearrangements from a robust number of cases from independent series of representative types of B cell malignancies is well documented and confirms by-and-large the results obtained on the same cases using Sanger sequencing of PCR-based Ig V gene amplicons. Data presented in this report also support the usefulness of IgCaller in predicting the IgH isotype expressed by the malignant B cells, as they were confirmed in representative cases by flow cytometric data. Finally, using the same set of WGS data, authors report in a subset of tumor cases the identification of chromosomal translocations targeting IgH/L chromosomes. The validation of such results using alternative molecular approaches, such as PCR amplification across the predicted chromosomal breakpoints awaits confirmation.

The manuscript by Nadeu and colleagues suffers from a lack of information on the power of IgCaller to identify rare tumor B cell clonal variants, including those possibly acquiring crippling Ig mutations in

the course of tumor progression and/or as a result of the exposure to specific treatments. The missing information is clinically relevant, as drugs targeting B cell receptor (BCR) signaling represent in many instances first-line therapies and premature recognition of clonal variants resisting to BCR therapies through the accumulation of crippling mutations within IgH/L V genes may benefit from early identification. The ability of IgCaller to identify rare clonal variants that have lost BCR expression, could be tested running the algorithm on WGS data from primary cases of Hodgkin Disease or normal germinal center B cells to search for crippling mutations targeting productive IgH/L V gene rearrangements.

The presented data fail to provide information on the ability of IgCaller to build Ig phylogenetic trees from malignant (and possibly also normal) germinal center B cells displaying ongoing somatic hypermutation, and therefore to reconstruct clonal B cell evolution in the course of tumor progression.

In summary, this brief report describes a versatile algorithm which complements a series of already existing ones aimed to reconstruct Ig V gene rearrangements from NGS short reads, to ultimately deconvolute clonal complexity in human mature B cell neoplasms. IgCaller has the advantage, over existing algorithms performing similar tasks, to couple IgV gene analysis with IgH isotype expression by the malignant B cells and to unravel at single nucleotide resolution the breakpoints of possible chromosomal translocations targeting IgH/L gene loci.

The manuscript is a clearly written methodological report, which lacks to provide in its present forms any new insights in the genetics of B cell lymphomas. At the same time, it misses critical information on the ability of IgCaller to track the evolution of malignant B cells during tumor progression, and possibly, also to study normal B cell repertoire dynamics in healthy and diseased patients.

IgCaller promises to become a useful tool for basic research in the field of Ig immunogenetics applied to B cell neoplasms. Its application in the clinical setting remains for the close future doubtful, as the exploitation of WGS (especially at sufficiently high coverage) for routine genetic analyses of B cell neoplasms remains a difficult task to achieve.

Specific points:

- 1) The manuscript fails to provide information on the frequency of tumor cases in which IgCaller identified two clonal IgH rearrangements, with one possibly representing an unproductive rearrangement. Similar information should be provided for IgL V gene rearrangements.
- 2) Information of WGS coverage data for each tumor analyzed should be included.
- 3) Figure 3a and 3c: data are poorly informative as flow cytometric data needed for the validation of the results are largely missing (i.e. C2-CLL and B-NHL for Figure 3A and C1-CLL, C2-CLL, MCL and B-NHL of Fig. 3c).
- 4) Suppl. Fig. 1b: what do the brown-labelled reads correspond to? Are they referring to a second clonal IgH V gene rearrangement (possibly unproductive) originating from the second IgH chromosome?
- 5) Suppl. Fig 1d: what do the red-labeled reads mapping downstream of IGHJ6 correspond to?
- 6) Suppl. Fig. 4: Reads annealing to heavily mutated regions of clonal IgH V gene rearrangements risk to be filtered out by IgCaller. Such limitation may hamper the recognition of malignant B cell variants possibly acquiring IgV gene crippling mutations on initially productive rearrangements. Have authors observed such behavior in tumors showing ongoing somatic hypermutation?
- 7) Suppl. Fig. 8b. The attempt to identify the IgH isotype expressed by the malignant B cells in the cases shown in Panel-c is doubtful. Is the difficulty to observe a drop in the coverage of the region downstream of the IgHM CSR linked to low coverage of WGS data of the specific cases and/or to a high frequency of contaminating non-tumoral polyclonal B cells?
- 8) Suppl. Fig. 9: including information on depth coverage respectively upstream and downstream of the IGHG1 CSR region would improve the clarity of the figure panel.

Point-by-point answers to Referee's comments

Reviewer #1 (BCR specificity, system immunology (BCR biased)) (Remarks to the Author):

Nadeu et al. present a data analysis approach for identifying the immunoglobulin gene sequences from whole genome sequence data of B cell-derived cancers. The method is generally very clearly explained, is freely available to other researchers, and is applied to a large set of published and unpublished genomes. This approach could be useful in the study of these cancers, particularly in cases where the Ig receptor may have a signaling role in pathogenesis. The current manuscript does not address the biological significance of the receptors identified, but the approach and software is likely to be of value to other researchers in the field.

We appreciate the global positive evaluation of our method. Our study was designed to characterize the IG rearrangements from WGS data. To that aim, we provided a free- and easy-to-use algorithm. The potential clinical and biological added value of IgCaller, beyond the characterization of the IG rearrangements, is highlighted by the discovery of oncogenic translocations missed by previous WGS analyses both in chronic lymphocytic leukemia (CLL) and diffuse large B-cell lymphoma (DLBCL) [now we included in the revised version of the manuscript the analysis of the WGS of 73 DLBCL cases], the identification of a subset of CLL with class-switched IG rearrangements that seem to have a more aggressive clinical behavior, and the confirmation of the clinical value of IGLV3-21 rearrangements in CLL. These later analyses are usually not included in conventional WGS studies.

Specific comments:

The authors do not address in much depth the effect that normal human variants in the Ig loci, such as copy number variants, will have on the IgCaller results. Were any of the alternative IGH loci sequence regions in the hg38 assembly tested in the analysis to see whether the results would be unaffected by alternative reference loci?

We agree with the Reviewer that we did not enter in much detail in this issue. Nonetheless, we show that IgCaller is able to identify secondary chromosomal alterations, including translocations, deletions, gains and inversions, involving the IG loci. We have not observed any effect on the accuracy of IgCaller on detecting V(D)J rearrangements in tumors carrying oncogenic IG alterations. We believe these results suggest that IgCaller is not influenced by copy number variants (or other structural events) within the IG loci. We have added this idea in page 8, line 195-197.

We have not specifically tested alternative IGH loci sequence regions in our analysis. However, as IgCaller works with WGS reads mapped using standard algorithms against the complete reference genome, we can conclude that the presence of alternative reference loci in the genome did not affected the sensitivity of IgCaller.

It would be important to determine the effect of the fraction of malignant clonal B cells present in a sample on the quality and accuracy of the IgCaller results. In samples that have significant numbers of non-malignant polyclonal B cells present in them, how is the IgCaller analysis affected?

We really appreciate this comment that highlights a relevant issue missing in the previous version of our manuscript. We have now added a new section in the results entitled "Effect of sequencing depth and tumor purity" in which we have addressed the effect of sequencing depth and levels of non-malignant or polyclonal B cell contamination in the tumor sample on the sensitivity of IgCaller. This new section is

accompanied by the new Figure 4, Supplementary Figure 12, Supplementary Table 24-28, and Online Methods “Downsampling and mixing of tumor and normal/polyclonal-like WGS data”. In short, the sensitivity of IgCaller is: i) relatively stable at low sequencing depths (10-15x); and ii) >0.8 at a tumor cell content of 20% (i.e. 80% of non-tumor contamination) when analyzing 60x WGS, while >0.8 at a tumor cell content of 50% for 30x WGS. Similarly, the sensitivity of IgCaller is >0.8 in the scenario of a 35% contamination of a polyclonal-like WGS sample created by mixing all 294 tumors. These analyses confirm that IgCaller is robust when moderate levels of non-tumor cells are mixed with the clonal tumor population. We think the decreased sensitivity of IgCaller at high levels of contamination is influenced by the low sequencing depth of WGS; an inherent bottleneck of this methodology. Nonetheless, we have also shown that IgCaller is able to characterize oligoclonal tumor populations.

The data disclosure should include the sequence reads from the currently unpublished cases whose Ig loci are used and analyzed in the study. To enable others to replicate the results, the reads from these genome regions should be deposited to a publicly-available repository (via dbGAP if required), rather than requiring readers to submit a request to the authors.

We have now uploaded all WGS reads in publicly-available repositories. Fifty-seven previously unpublished MCL cases has been deposited in the European Genome-phenome Archive (EGA) under accession EGAS00001004165. The IG reads for C2-CLL and B-NHL cohorts as well as 3 MCL cases have been deposited in the EGA under accession number EGAS00001004298. We have updated the “Data Availability” section of the manuscript accordingly.

Reviewer #2 (B response, leukemia, cancer) (Remarks to the Author):

The manuscript „IgCaller: Reconstructing the Immunoglobulin Gene Rearrangements and Oncogenic Translocations in Lymphoid Neoplasms from Whole-Genome Sequencing” by Nadeu et al. reports an algorithm designated IgCaller which uses short-read WGS (whole-genome sequence) data to characterize immunoglobulin (Ig) gene rearrangements, sequence alterations and oncogenic translocations.

Generally, the VDJ gene rearrangement is central for determining the clonality of a B cell population in lymphoid neoplasms whereby individual VDJs serve as markers for the diagnosis of lymphoid neoplasms. Moreover, the mutational status of Ig genes is associated with specific clinical outcome in lymphoid malignancies such as chronic lymphocytic leukemia (CLL) or mantle cell lymphoma (MCL). Therefore, the identification of Ig gene mutations as compared with reference germ line sequence is an important step in the diagnosis of CLL for instance.

I found the paper interesting and I can appreciate the usefulness of the IgCaller for clinicians or scientists who study lymphoid malignancies and routinely use WGS. However, the manuscript describes a method and lacks clear novel finding. For instance, what is the addition to the current knowledge about B cell malignancies? Also, what is surprising is the conclusion that the presence of class switch recombination (CSR) in M-IGHV (16 of 106 cases or 15%) “identified CLL patients with a tendency to a shorter time to first treatment (TTFT) than non-switched M-IGHV CLL (Fig. 3d)”. However, stereotype subset#4 CLL is mutated, expresses IgG (thus switched) and shows a very indolent course of disease as it mostly does not require therapy. This is in sharp contrast to the authors’ conclusion.

We appreciate the Referee found our algorithm useful for the community. We think the novelty of our study relies on being the first algorithm to allow the complete reconstruction of the IG rearrangements from short-read, low-coverage WGS. We believe our approach might assist the characterization of the IG rearrangements in further studies of B-cell neoplasms, which indeed might contribute to better characterize their biological and clinical relevance. We think that the use of IgCaller in the analysis of the WGS of B-cell neoplasm may better characterized the BCR providing relevant biological and clinical information that is missed by current pipelines. Beyond the complete structural characterization of the IG rearrangements, IgCaller has discovered oncogenic translocations missed by previous WGS analyses both in chronic lymphocytic leukemia (CLL) and diffuse large B-cell lymphoma (DLBCL) [now included in the revised version of the manuscript 73 WGS of DLBCL], it has identified a subset of CLL with class-switched IG rearrangements that seem to have a more aggressive clinical behavior, and has confirmed the clinical value of the specific IGLV3-21 rearrangements in a subset of CLL. These later analyses are usually not included in conventional WGS analyses.

We agree with this Referee that specific CLL stereotypes have marked distinct clinical significance. As pointed out by his/her comment, subset #4, and probably also #16 (Xochelli *et al*, Clin Cancer Res. 2017;23(17):5292-5301), although class switched, have a very indolent course compared to other CLL with mutated IGHV or CLL carrying a non-stereotyped V4-34 rearrangement. In our study, we did not have any case with these stereotypes. We compared IgG switched and non-switched CLL with mutated IGHV and showed that IgG switched cases had a tendency to shorter time to first treatment. We have now amended this sentence in the main text to clarify this point (page 7, lines 175-178). We would like to emphasize here that the analysis of stereotypy is also possible with IgCaller based on the full-length characterization of the V(D)J rearrangement in most of the cases analyzed. We have highlighted this idea (page 5, lines 113-115) and added the stereotype of each CLL cases in the new version of Supplementary Tables 2-3. Altogether, we think that the full characterization of the IG loci using IgCaller might help distilling subtle but clinically relevant conclusions as the one pointed out by this Referee.

Reviewer #3 (B cell biology, BCR, VDJ rearrangement) (Remarks to the Author):

In this report, Nadeu and colleagues describe IgCaller, a versatile bioinformatic tool aimed to reconstruct the identity of clonal Immunoglobulin (Ig) heavy (H) and light (L) chain V gene rearrangements from short sequencing reads obtained from whole genome sequencing (WGS) of primary specimens of human B cell malignancies.

The Ig caller algorithm offers the opportunity to combine Ig VDJ profiling with the identification of the possible IgH isotype expressed by the clonal malignant B cell population, and to the mapping at nucleotide resolution of the breakpoint of possible chromosomal translocations involving IgH/L chain gene loci.

IgCaller is the latest of a series of algorithms developed in recent years to extrapolate Ig VH/VL gene repertoires from healthy and/or pathological B cell populations using Next Generation Sequencing (NGS)-based approaches. IgCaller complements current algorithms as it exploits short sequencing reads generated by WGS. Such practice is yet rarely employed to identify the identity of Ig V gene clonotypes defining pathological B cell population(s) contributing to disorders such as B cell neoplasms.

IgCaller exploits two major features to reconstruct unique V(D)J joining regions, namely so-called split reads and abnormal insert size reads, after annealing short reads coming from WGS to enriched genomic regions covering IgH and IgL chain loci. Such principle has been previously employed by other Ig repertoire reconstruction algorithms, which are centered on the analysis of NGS reads obtained from Ig V gene targeted and whole exome sequencing datasets. The present manuscript lacks any form of benchmarking of IgCaller with currently available algorithms reconstructing IgV gene rearrangements from NGS short reads.

We appreciate the positive comments of the Referee and thank him/her for their useful suggestions. We would like to emphasize here that IgCaller does not perform any enrichment neither any specific alignment of WGS reads. It works with short-read, low coverage WGS data aligned using *standard* algorithms. The same aligned reads (i.e. BAM file) can be used (and indeed were used in previous studies) to extract mutations, copy number alterations or structural variants, among other information. The unique “preprocessing” step performed by IgCaller is to retain only those reads mapping within the IG loci (using samtools view) to facilitate a faster run than if iterating the whole genome. We have emphasized these ideas along the manuscript, especially in the first paragraph of the discussion (page 11, lines 277-280).

After a thorough search, we were unable to find any study that claimed that IG gene rearrangements could be extracted from WGS. Nonetheless, we have now benchmarked IgCaller with MiXCR (Bolotin *et al*, Nat Methods 2015 2015;12(5):380-381), which we think is a well-accepted algorithm for the profiling of the IG repertoire from RNA-seq data and the authors proposed that the algorithm should work for any type of single- and paired-end sequencing data (both RNA and DNA). We have added a new section in the manuscript entitled “Comparison of IgCaller with other available algorithms” which summarizes the results. Briefly, although MiXCR identified the same clone (based on CDR3 sequence) for 194 samples tested and the same V(D)J gene in 174/194 (90%) cases, it failed to reconstruct the entire V(D)J region impairing a proper analysis of the IGHV identity of the sequences. This discrepancy might be explained by the different approach of both algorithms. See new text in page 10 of the manuscript and new Supplementary Table 29.

The present manuscript lacks information on the minimum WGS coverage required to faithfully reconstruct IgVH/VL gene clonotypes representative of malignant B cells within the tumor biopsy. It remains also undetermined the performance of the algorithm under conditions where putative clonal B cell population(s) in a starting tumor are overwhelmed by a majority of polyclonal wild-type B cells. Testing the performance of the algorithm under controlled conditions where WGS data are produced from a unique tumor population (i.e. a tumor cell line) mixed at different ratios with a polyclonal B cell population may provide useful information on the sensitivity of an approach that may suffer from a possible low coverage of the relevant genomic data.

We thank the Reviewer for this very relevant point and agree that the manuscript lacked important information regarding the minimum sequencing depth and tumor cell content to properly run IgCaller. We have now addressed these issues in the new section “Effect of sequencing depth and tumor purity” (page 8-9), which is accompanied by new Figure 4, Supplementary Figure 12, Supplementary Tables 24-28, and Online Methods “Downsampling and mixing of tumor and normal/polyclonal-like WGS data”.

Briefly, to study the effect of coverage:

- We have included the mean depth of coverage of the IG loci for each sample in the new version of Supplementary Tables 2-7. Note that mean coverages ranged mainly between 30 and 75x.
- We have analyzed if there was an effect on the sensitivity of IgCaller at different coverage in the studied CLL and MCL cohorts (Fig. 4a).
- We have downsampled the WGS of 29 samples with tumor cell content >90% at different mean coverages (30x, 25x, 20x, 15x, 10x, and 5x) and analyzed the sensitivity on the reconstruction of the IGH V(D)J gene rearrangement and IGHV gene identity (Fig. 4b-c).

To summarize the results, the sensitivity to detect the productive IGH and IGK/L gene rearrangements was close to 0.9 and 0.75 at 15 and 10x, respectively. The IGHV gene identity was minimally affected by sequencing depths ranging from 15x to 30x. As expected, some mutations were missed at very low coverages (5-10x).

To study the effect of the tumor cell content in the output of IgCaller:

- We mixed the tumor and non-tumoral WGS reads from 29 paired samples at different ratios (final tumor cell content of 5%, 20%, 35%, 50%, 65%, 80%, and 95% -starting point) and at a final sequencing depth of 30x and 60x. The sensitivity of IgCaller was >0.8 at a tumor cell content of 20% (i.e. 80% of non-tumor contamination) when analyzing 60x WGS, while >0.8 at a tumor cell content of 50% for 30x WGS (Fig. 4d-e). The IGHV gene identity was minimally affected by normal contamination of the tumor in mixed samples with >35% tumor cell content (Fig. 4f).
- To run IgCaller in a polyclonal setting, we have created a polyclonal-like B cell population in silico by mixing all 294 CLL and MCL tumor samples. We then mixed at different ratios the 29 previously used tumor samples with purities >90% with this polyclonal sample. The effect of polyclonal B-cell contamination was similar to that observed for normal, non-B-cell contamination (Fig. 4g, Supplementary Figure 12).
- Finally, we also analyzed the sensitivity of IgCaller to characterize oligoclonal samples by mixing two C1-CLL tumors carrying both a productive and unproductive IGH gene rearrangement (Fig. 4h).

We have also commented on these results in the Discussion (page 11-12).

The analyses, using IgCaller, of IgH/L V gene rearrangements from a robust number of cases from independent series of representative types of B cell malignancies is well documented and confirms by-and-large the results obtained on the same cases using Sanger sequencing of PCR-based Ig V gene amplicons. Data presented in this report also support the usefulness of IgCaller in predicting the IgH isotype expressed by the malignant B cells, as they were confirmed in representative cases by flow cytometric data. Finally, using the same set of WGS data, authors report in a subset of tumor cases the identification of chromosomal translocations targeting IgH/L chromosomes. The validation of such results using alternative molecular approaches, such as PCR amplification across the predicted chromosomal breakpoints awaits confirmation.

We appreciate the positive comment of the Referee. Regarding the validation of the chromosomal translocations identified, we would like to clarify that all the alterations found in 54 selected cases were verified by FISH, cytogenetics and/or PCR (Supplementary Table 23). We have better explained this verification process on page 8, line 183-184.

The manuscript by Nadeu and colleagues suffers from a lack of information on the power of IgCaller to identify rare tumor B cell clonal variants, including those possibly acquiring crippling Ig mutations in the course of tumor progression and/or as a result of the exposure to specific treatments. The missing information is clinically relevant, as drugs targeting B cell receptor (BCR) signaling represent in many instances first-line therapies and premature recognition of clonal variants resisting to BCR therapies through the accumulation of crippling mutations within IgH/L V genes may benefit from early identification. The ability of IgCaller to identify rare clonal variants that have lost BCR expression, could be tested running the algorithm on WGS data from primary cases of Hodgkin Disease or normal germinal center B cells to search for crippling mutations targeting productive IgH/L V gene rearrangements.

We thank the Referee for highlighting the possibility of IgCaller to detect crippling mutations targeting IgH/L V rearrangements. Although it is of interest, we did not focus on the ability of IgCaller to identify rare tumor B cell clonal variants due to the limitation of low coverage WGS to characterize minor subclonal populations (i.e. minor subclonal populations undergoing SHM). Regarding the idea of studying Hodgkin lymphoma, we think that the low tumor cell content in primary cases of this lymphoma (usually <10%) impairs its analysis by WGS. We are not aware of published WGS of this tumor. Nonetheless, to address the issue of the detection of crippling mutations raised by the reviewer, we have now specifically searched for the presence of crippling mutations in our cases. Among the 51 non-productive IGH rearrangements identified by IgCaller, 44 carried mutations leading to stop codons, confirming that these types of mutations can be detected with our method. We have now included this information in the new version of Supplementary Table 8. IgCaller also identified unproductive IGK/L gene rearrangements carrying mutations leading to stop codons (see new version of Supplementary Table 16).

To address whether IgCaller detect subclonal crippling mutations we have specifically searched for these mutations in the sensitivity experiment described in the previous point (Fig. 4h). In this new analysis, IgCaller has been able to detect the unproductive rearrangements due to crippling mutations when present in around 25% of the tumor cell content.

The presented data fail to provide information on the ability of IgCaller to build Ig phylogenetic trees from malignant (and possibly also normal) germinal center B cells displaying ongoing somatic hypermutation, and therefore to reconstruct clonal B cell evolution in the course of tumor progression.

The point raised by the Referee is of interest but we agree that whole genome sequences of tumor or normal tissues at conventional coverage may not be the most appropriate method to build Ig phylogenetic trees. In our WGS data with mostly 30x-40x of mean sequencing coverage, clonal mutations

are detected with approximately 15-20 reads. Considering sequencing bias on specific regions, unequal representation of the alleles, and the lower mappability rate of reads spanning structural rearrangements, clonal mutations within the IG genes are usually detected at lower frequencies. Based on that, the possibility to robustly identify very subclonal IG populations and properly extract phylogenetic trees is beyond the resolution of the method. We have discussed this limitation in the Discussion (page 12) and pointed to other sequencing techniques (Rep-Seq or RNA-seq) and tools (MiXCR) for those interested in these aspects.

In summary, this brief report describes a versatile algorithm which complements a series of already existing ones aimed to reconstruct Ig V gene rearrangements from NGS short reads, to ultimately deconvolute clonal complexity in human mature B cell neoplasms. IgCaller has the advantage, over existing algorithms performing similar tasks, to couple IgV gene analysis with IgH isotype expression by the malignant B cells and to unravel at single nucleotide resolution the breakpoints of possible chromosomal translocations targeting IgH/L gene loci.

The manuscript is a clearly written methodological report, which lacks to provide in its present forms any new insights in the genetics of B cell lymphomas. At the same time, it misses critical information on the ability of IgCaller to track the evolution of malignant B cells during tumor progression, and possibly, also to study normal B cell repertoire dynamics in healthy and diseased patients.

IgCaller promises to become a useful tool for basic research in the field of Ig immunogenetics applied to B cell neoplasms. Its application in the clinical setting remains for the close future doubtful, as the exploitation of WGS (especially at sufficiently high coverage) for routine genetic analyses of B cell neoplasms remains a difficult task to achieve.

We appreciate the positive global evaluation of our work. The aim of our study was to demonstrate that the complete IG gene might be reconstructed from short-read, low coverage WGS that until now has not been feasible with the current bioinformatic tools. WGS is already used for the diagnosis of specific diseases in the clinics in some countries. We and others have shown that the genomic landscape of these tumors can be very well characterized from low coverage WGS data and it has been proposed that this approach may be useful for clinical trials, drug discovery, and precision medicine (Klintman *et al*, Br J Haematol 2018;182(3):412-417). As proposed, this low coverage WGS might well substitute, for instance, the use of FISH, cytogenetics and Sanger sequencing performed in the daily diagnosis of CLL. High coverage WGS would be needed to tackle ongoing SHM or to properly study phylogenies within tumor and/or normal B cell populations. We are not aware that these issues, although of biological relevance, might impact the diagnosis of these diseases neither the management of the patients. We believe IgCaller may assist in both scenarios thanks to the identification of clinically relevant oncogenic IG rearrangements as well as the full V(D)J gene sequence to properly characterize their IGHV identity. These factors are both prognostic and predictive in all the diseases studied here.

Specific points:

- 1) The manuscript fails to provide information on the frequency of tumor cases in which IgCaller identified two clonal IgH rearrangements, with one possibly representing an unproductive rearrangement. Similar information should be provided for IGL V gene rearrangements.

We thank the Referee for highlighting this important aspect. We have now added this information in the new version of Supplementary Table 8 (unproductive IGH rearrangements) and Supplementary Table 16 (unproductive IGL rearrangements). We have also further discussed the ability of IgCaller to identify both productive and unproductive rearrangement in the main text: page 5, line 116-118 for IGH; page 6-7, line

149-155 for IGL. As mentioned above, these results also emphasize the sensitivity of IgCaller to detect multiple rearrangements within a tumor sample as well as gene rearrangements carrying clipping mutations.

2) Information of WGS coverage data for each tumor analyzed should be included.

We have included the mean coverage of the IG loci for each sample in the new versions of Supplementary Tables 2-7.

3) Figure 3a and 3c: data are poorly informative as flow cytometric data needed for the validation of the results are largely missing (i.e. C2-CLL and B-NHL for Figure 3A and C1-CLL, C2-CLL, MCL and B-NHL of Fig. 3c).

We agree these panels are not extremely informative from the point of view of the validation based on flow cytometry data. Nonetheless, we think the representation of the CSR in both CLL cohorts, although lacking flow cytometry data, visually shows the percentage of cases that undergone CSR. The similar frequency of switched cases in both CLL cohorts argues in favor of the robustness of IgCaller in terms of the detection of CSR observed in MM (with flow cytometry available). From a biological perspective, we think that it might be of interest to some readers to visually observe the remarkable difference between CLL and MCL in terms of CSR. In this sense, our study shows the potential prognostic value of CSR in CLL with mutated IG. Besides, the distribution of CSR observed in the different tumor types is similar to that observed using flow cytometry or Sanger sequencing in previously studied cohorts. We have added this idea on page 7, lines 170-171.

4) Suppl. Fig. 1b: what do the brown-labelled reads correspond to? Are they referring to a second clonal IgH V gene rearrangement (possibly unproductive) originating from the second IgH chromosome?

Brown-labeled reads in Supplementary Figure 1b correspond to the reads spanning the t(11;14) (IG/CCND1) present in this MCL. More precisely, these reads are labelled in brown because their mate align near *CCND1* on chromosome 11. Note this t(11;14) is also identified by IgCaller. We have added this explanation in the figure legend.

5) Suppl. Fig 1d: what do the red-labeled reads mapping downstream of IGHJ6 correspond to?

Red-labeled reads mapping downstream of IGHJ6 correspond to a deletion on the non-V(D)J rearranged allele with the second breakpoint found between the IGHD6-19 and IGHD5-18 genes. Obviously, these reads are not considered for the analysis of the V(D)J productive rearrangement found in the other allele. We thank the reviewer for highlighting that this explanation was missing. It has been now added to the legend of Supplementary Figure 1D.

6) Suppl. Fig. 4: Reads annealing to heavily mutated regions of clonal IgH V gene rearrangements risk to be filtered out by IgCaller. Such limitation may hamper the recognition of malignant B cell variants possibly acquiring IgV gene crippling mutations on initially productive rearrangements. Have authors observed such behavior in tumors showing ongoing somatic hypermutation?

As discussed before, IgCaller is able to identify unproductive gene rearrangements carrying crippling mutations even if present in around 25% of the tumor sample. To better analyze the effect of heavily mutated regions and potential ongoing somatic hypermutation in the output of IgCaller we have now included 73 diffuse large B-cell lymphoma (DLBCL) (31 activated B-cell type -ABC-; 33 germinal center B-cell type -GCB-, and 9 unclassified). GCB-DLBCL, but not ABC-DLBCL, carry ongoing somatic mutations

(Lossos *et al*, Proc Natl Acad Sci USA 2000;97(18):10209-13). IgCaller identified a productive IGH gene rearrangement in 63% of cases with a similar distribution in ABC and GCB subtypes (21, 68% ABC; and 18, 55% GCB) and IGK/L in 62% (61% ABC, 64% GCB). These results are similar to those observed with traditional Sanger sequencing using well-established protocols (Evans *et al*, Leukemia 2007;21(2):207-14). The use of this DLBCL cohort has also allowed us to prove the ability of IgCaller on identifying oncogenic IG translocations in this lymphoma, with 3 translocations not identified in the previous WGS analysis (see Page 8, section 'Oncogenic IG translocations').

7) Suppl. Fig. 8b. The attempt to identify the IgH isotype expressed by the malignant B cells in the cases shown in Panel-c is doubtful. Is the difficulty to observe a drop in the coverage of the region downstream of the IgHM CSR linked to low coverage of WGS data of the specific cases and/or to a high frequency of contaminating non-tumoral polyclonal B cells?

We are not sure why in this discordant case we are not able to identify the CSR matching the isotype observed using flow cytometry. The tumor purity of the sample is high (86%) and the sequencing depth is similar to all other MM samples (31.41x of mean coverage). In this sample we cannot identify a productive IGH gene rearrangement neither, but successfully characterized the IG kappa rearrangement. Said that, we cannot give a conclusive explanation why we do not see a drop of coverage in this IgG MM according to flow cytometry.

8) Suppl. Fig. 9: including information on depth coverage respectively upstream and downstream of the IGHG1 CSR region would improve the clarity of the figure panel.

We have added the mean depth of coverage of both regions. We agree this has improved the clarity of the figure. Thank you for the suggestion.

REVIEWERS' COMMENTS:

Reviewer #3 (Remarks to the Author):

The revised manuscript by Nadeu and colleagues has addressed the main concerns of the reviewer. The revised manuscript has improved its quality through the inclusion of further datasets of lymphoma whole genome sequencing (WGS), and improved analyses of previous results. Authors are recommended to render freely available all WGS datasets that were presented in order to ensure reproducibility of the results by the scientific community.

The revised manuscript offers minor novel biological insights related to the pathogenesis of lymphomas. However, this lack is balanced by the description of a novel informatics tool, that is predicted to be of great value for the scientific community working in the field of lymphoma/leukemia genetics. It promises also to help in the clinical setting to define/refine lymphoma clonal identity and Ig-associated chromosomal rearrangements, with potential diagnostic and prognostic implications.

Point-by-point response to Reviewers' comments (response in red)

Reviewer #3 (Remarks to the Author):

The revised manuscript by Nadeu and colleagues has addressed the main concerns of the reviewer. The revised manuscript has improved its quality through the inclusion of further datasets of lymphoma whole genome sequencing (WGS), and improved analyses of previous results. Authors are recommended to render freely available all WGS datasets that were presented in order to ensure reproducibility of the results by the scientific community.

The revised manuscript offers minor novel biological insights related to the pathogenesis of lymphomas. However, this lack is balanced by the description of a novel informatics tool, that is predicted to be of great value for the scientific community working in the field of lymphoma/leukemia genetics. It promises also to help in the clinical setting to define/refine lymphoma clonal identity and Ig-associated chromosomal rearrangements, with potential diagnostic and prognostic implications.

We appreciate the positive comments of the reviewer. All whole-genome sequencing data used along the manuscript is now deposited on EGA. We have updated the Data Availability section of the manuscript accordingly.